# Borax induces osteogenesis by stimulating NaBC1 transporter via activation of BMP pathway

Patricia Rico [1,2✉], Aleixandre Rodrigo-Navarro[3], Laura Sánchez Pérez[2] & Manuel Salmeron-Sanchez [1,2,3✉]

The intrinsic properties of mesenchymal stem cells (MSCs) make them ideal candidates for tissue engineering applications. Efforts have been made to control MSC behavior by using material systems to engineer synthetic extracellular matrices and/or include soluble factors in the media. This work proposes a simple approach based on ion transporter stimulation to determine stem cell fate that avoids the use of growth factors. Addition of borax alone, transported by the NaBC1-transporter, enhanced MSC adhesion and contractility, promoted osteogenesis and inhibited adipogenesis. Stimulated-NaBC1 promoted osteogenesis via the BMP canonical pathway (comprising Smad1/YAP nucleus translocation and osteopontin expression) through a mechanism that involves simultaneous NaBC1/BMPR1A and NaBC1/$\alpha_5\beta_1/\alpha_v\beta_3$ co-localization. We describe an original function for NaBC1 transporter, besides controlling borate homeostasis, capable of stimulating growth factor receptors and fibronectin-binding integrins. Our results open up new biomaterial engineering approaches for biomedical applications by a cost-effective strategy that avoids the use of soluble growth factors.

---

[1] Biomedical Research Networking Center in Bioengineering, Biomaterials and Nanomedicine (CIBER-BBN), 28029 Madrid, Spain. [2] Centre for Biomaterials and Tissue Engineering (CBIT), Universitat Politècnica de València, Camino de Vera s/n, 46022 Valencia, Spain. [3] Centre for the Cellular Microenvironment, University of Glasgow, Glasgow G12 8LT, UK. ✉email: parico@upvnet.upv.es; Manuel.Salmeron-Sanchez@glasgow.ac.uk

Mesenchymal stem cells (MSCs) are multipotent and capable of differentiating mesodermal lineages (reticular, adipogenic, osteogenic and chondrogenic) under certain conditions that often include growth factors[1]. Their final fate in vivo depends on a combination of physical, chemical and biological cues, all of which are present in the natural stem cell niche. The extracellular matrix (ECM) of the stem cell niche plays an important role by supporting cell growth, dynamically regulating growth factor release and activating multiple intracellular pathways[2,3]. Material systems offer an alternative for accurately engineering synthetic ECMs. The current material-based strategies for directing stem cell behavior are based on varying the material properties in terms of chemistry[4], stiffness[5,6], or topography[7,8]. Other approaches use growth factors dissolved in the media, delivered from material systems or as components of bioactive materials for efficient solid-phase presentation[9]. However, although they are widely used clinically in high concentrations, they can also produce undesired side effects such as neurological problems or cancer[10], so that alternative methods are needed to avoid supraphysiological doses of growth factors.

Among the many factors governing MSCs commitment, mechanical cues have emerged as key determinants in cell fate and function in a context-dependent manner[11]. Force exerted by the cells on the ECM is mediated by major adhesion receptors, integrins, mechanosensors that pull on ECM proteins and transmit mechanical forces. This interaction causes cytoskeleton contraction and integrin clustering, giving rise to focal adhesions[12] and an intracellular cascade of downstream signal transduction events that determine the stem cell fate. It is widely accepted that the osteogenic/adipogenic balance is determined by the initial MSC adhesion: i.e., poorly adhered MSCs (small focal adhesions) differentiate into rounded adipocytes[7,13] while MSCs that develop mature adhesions (long in size, allowing cell spreading and intracellular tension) differentiate into osteoblasts[6,13]. Besides integrins, cells can also perceive force through ion channels. This activation occurs after ligand binding, membrane stretch, interaction with other specific ligands or changes in membrane potential[14], so that ion channels can act as mechanosensors that communicate extracellular signals to the cytoplasmic environment[15] and integrins[16]. The contribution of ion channels to the regulation of cell behavior in MSCs is becoming increasingly recognized, although the precise mechanisms are still debated and the information available mainly refers to the $Ca^{2+}$, $K^+$, and $Cl^-$ channels[17].

The NaBC1 transporter controls boron (B) homeostasis and, in the presence of borate, functions as an obligated $Na^+$-coupled borate co-transporter[18]. Mutations in the NaBC1 gene cause endothelial corneal dystrophies[19,20] and are overexpressed in most breast cancer cell lines and downregulated in some colorectal tumors[18]. We previously reported that B promotes myogenic differentiation[21], proposed a 3D molecular model for NaBC1 and showed that the simultaneous stimulation of NaBC1 and the vascular endothelial growth factor receptor (VEGFR) promote angiogenesis in vitro and in vivo with ultralow doses of growth factors[22]. Although little is known about borate homeostasis and function in mammalian cells, there are several reports that describe the role of this metalloid enhancing MSC osteogenic differentiation[23–25], and only recently others have described the inhibition of adipogenic differentiation[26,27].

Here, we report that addition of borax to the culture media induces osteogenesis and inhibits adipogenesis in the absence of other soluble chemicals or growth factors. We propose a new mechanism involving crosstalk and co-localization of active NaBC1/BMPR1A and NaBC1/$\alpha_5\beta_1$/$\alpha_v\beta_3$ integrins that activate intracellular pathways, with NaBC1 in the novel function of stimulating growth factor receptors and fibronectin-binding integrins.

## Results

With the aim of evaluating MSC osteogenic commitment, we used a combined system for simultaneous stimulation of NaBC1 and $\alpha_5\beta_1$ and $\alpha_v\beta_3$ integrins[28]. Boron (B) in the form of Sodium Tetraborate Decahydrate (borax) was used dissolved in the culture media for NaBC1 stimulation, and fibronectin (FN) for FN-binding integrin activation. We used Glass (control substrate) and polylactic acid (PLLA) as a possible FDA approved[29] biodegradable substrate for B delivery in future in vivo applications.

We used C3H10T1/2 cells, a murine multipotent MSC line from mouse embryo as the model cellular system, widely used in the study of cell differentiation mechanisms and capable of differentiation into mesodermal lineages (reticular, adipogenic, osteogenic and chondrogenic)[30,31].

We first evaluated borax cytoxicity to assess the maximum working concentrations for this particular cell line. Supplementary Fig. S1 shows the viability results obtained, indicating that borax concentrations higher than 26.2 mM (10 mg mL$^{-1}$) are toxic for C3H10T1/2 cells. In all the experiments, we supplemented the culture media with 0.59 mM (labeled B2%) and 1.47 mM (labeled B5%) from a borax solution, concentrations within the 0.2–0.6 mg mL$^{-1}$ range, to avoid any borax toxic effects on cells.

**NaBC1 stimulation induces MSCs adhesion.** All the substrates employed were coated with a 20 μg mL$^{-1}$ concentration of human plasma FN solution. We previously reported that FN adsorbs similarly on Glass and PLLA substrates in the presence/absence of borax, disregarding any influence of the ion on FN surface density[21].

We evaluated the combined effect of stimulating NaBC1 together with FN-binding integrins after 3 h of culture using serum-depleted culture media to ensure that initial cell-material interaction was exclusively FN-mediated, and so targeting $\alpha_5\beta_1$ and $\alpha_v\beta_3$ integrins[28]. Cells were seeded at low density (5000 cells cm$^{-2}$) to favor cell-material interaction and minimize cell-cell contacts in order to quantify focal adhesions (FA). We have performed the experiments using intact (non-transfected cells), transfected MSCs with esiRNA$^{NC}$ (fluorescent-labeled universal negative control with no sequence homology to any known gene sequence) and esiRNA$^{NaBC1}$ (with specific NaBC1 sequence homology).

We first assessed the transfection efficiency of the silencing experiment, detecting a clear red fluorescence in MSC cells after 24 h and 3 days post-transfection (Supplementary Fig. S2-a). Real time quantitative PCR (qPCR) amplification resulted in a decrease of NaBC1 mRNA levels after transfecting cells with esiRNA$^{NaBC1}$, confirming a successful silencing of the transporter (Supplementary Fig. S2-b).

Considering the stiffness range of the Glass (50–90 GPa)[32] and PLLA (3.5 GPa)[33] used as substrates, and the range of force that cells can exert (up to 5 nN μm$^{-2}$)[34], we hypothesized that the cells would detect all the substrates as rigid and so no result can be attributed to differences in the mechanical sensing of the environment. Figure 1a shows MSC morphology before and after NaBC1 silencing. Cells displayed marked actin stress fibers ending at well-developed focal adhesion sites of attachment that were more evident in the PLLA-B2% and PLLA-B5% substrates in non-transfected and esiRNA$^{NC}$ cells. Quantification of cell parameters showed that the presence of borax induced significant differences in the cell spreading area, even though the same number of cells was used in all the substrates (Supplementary Fig. S2-c).

FA quantification gave a higher total FA number and larger FA area in the presence of borax (Fig. 1b). After NaBC1 silencing, we

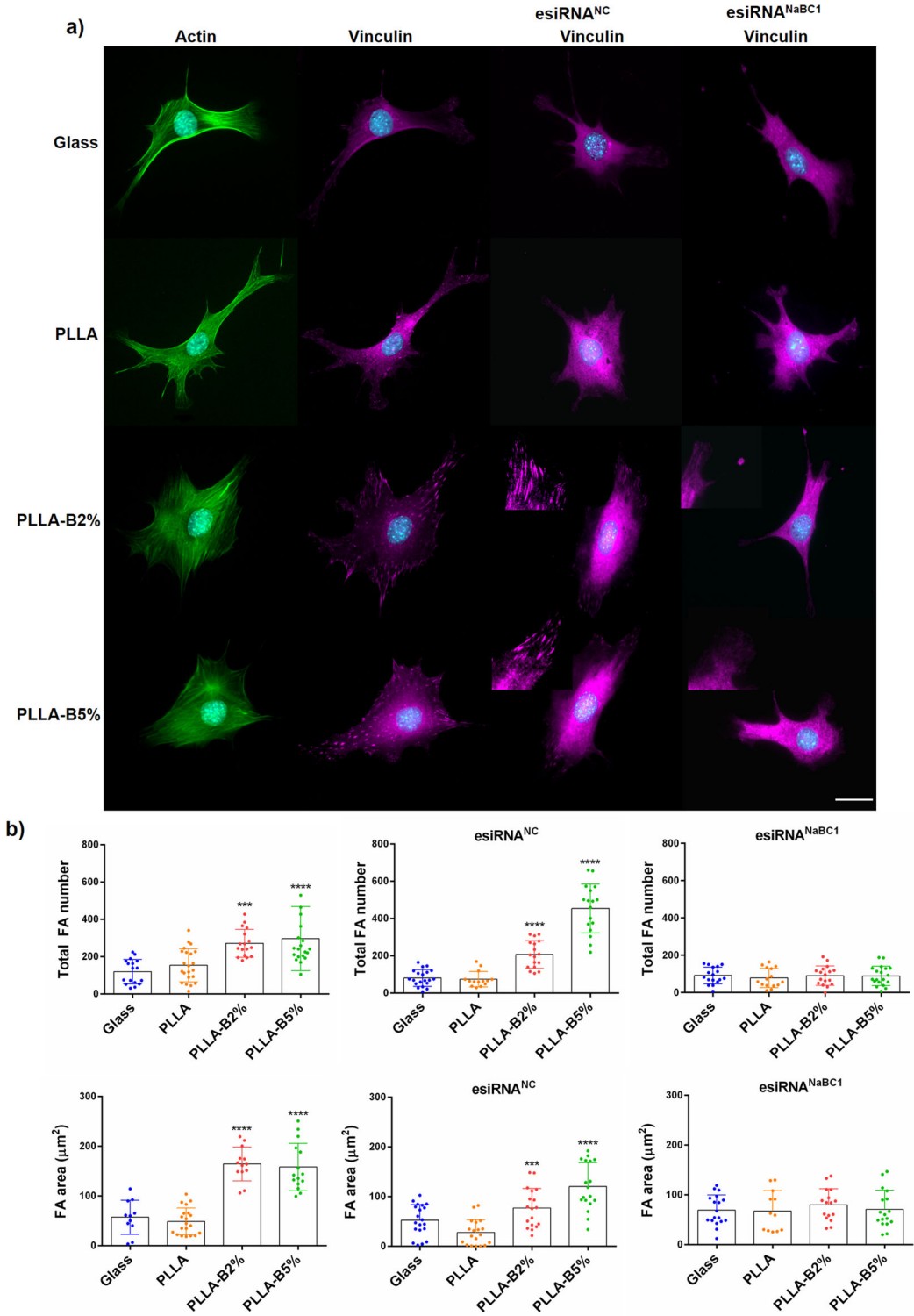

**Fig. 1 Borax effects on Focal Adhesion (FA) formation and MSC adhesion. a** Immunofluorescence images of actin cytoskeleton (green), nuclei (cyan) and vinculin (magenta) as a marker of focal adhesions (FA). MSCs were cultured for 3 h, onto functionalized substrates (FN-coated), with serum-depleted and borax (PLLA-B2%, PLLA-B5%) presence in the culture medium. The same experiment was performed after NaBC1 silencing using esiRNAs (esiRNA^NaBC1). Universal negative control esiRNA (esiRNA^NC) was used as a transfection efficiency control. The PLLA-B2% and PLLA-B5% substrates presented clear focal adhesions as a result of vinculin staining that strongly diminished after NaBC1 silencing (see inset magnifications). Scale bar 25 μm. **b** Image analysis quantification of different parameters related to FA. Total FA number and FA area before and after NaBC1 silencing. Borax induces more and bigger FAs, effect that is reverted after NaBC1 silencing. Twenty images/condition from three different biological replicas were analyzed. The data represented in graphs correspond to $n \geq 11$. Statistics are shown as mean ± standard deviation. Data were analyzed by an ordinary one-way ANOVA test and corrected for multiple comparisons using Dunnett analysis ($P = 0.05$). ***$p < 0.001$, ****$p < 0.0001$.

observed a strong decrease in the total FA number and area, even in the presence of borax (Fig. 1a, b), indicative that the enhancement of MSC adhesion depends on a functional NaBC1. The FA distribution frequency was similar on all the surfaces, with a greater number of nascent plaques (between 0 and 6 μm), which decreased monotonically until they became mature plaques (>6 μm). A detailed analysis of FA distribution revealed that borax induced FA formation at different FA development stages, resulting in higher levels of nascent (0–6 μm) and mature FA (6–12 μm) (Supplementary Fig. S2-d). Since integrins initiate the adhesion process by generating nascent integrin-matrix linkages that afterwards develop into mature integrin-ECM linkages, recruiting additional components under force[35], our results indicate that active NaBC1 accelerates integrin clustering to form FA in the early stages that will become mature FA in PLLA-B2% and PLLA-B5% substrates, and demonstrates that NaBC1 activation regulates cell adhesion as previously reported for other ion channels[36].

We next examined whether cell adhesion triggered by borax activation of NaBC1 involved activation (phosphorylation in the Tyr 397) of Focal Adhesion Kinase (FAK) as one of the main adapter proteins composing the mechanosensitive adhesome, which can directly regulate its catalytic activity through force-induced structural rearrangements[37]. In-Cell Western results showed no significant effects on FAK phosphorylation after the simultaneous stimulation of the FN-binding integrins and NaBC1 transporter (Supplementary Fig. S2-e), suggesting that the borax-induced mode of action of cell adhesion takes place via different pathways.

**Borax upregulates NaBC1 and FN-binding integrins expression in MSC.** The sodium-borate co-transporter NaBC1, essential for borate homeostasis, has already been characterized[18]. We recently reported a 3D molecular model based on the sequence homology of other bicarbonate transporter-related proteins[22]. The NaBC1 transporter is borate-specific and functions as an obligated $Na^+$-coupled borate co-transporter. As we used Sodium Tetraborate Decahydrate (borax) in this work, we confirmed that borax is the natural ligand for NaBC1 activation and that its transport is not through other mechanisms, such as membrane passive diffusion, as with boric acid[18].

Although there is some evidence for the tissue ubiquity of NaBC1[38], we confirmed its presence in our cellular system under the same experimental conditions (after 3 or 15 days of culture and under basal or differentiation media) used in subsequent differentiation studies. When MSCs NaBC1 mRNA were amplified by qPCR, the results showed that in all the conditions assessed, *NaBC1* gene expression was upregulated exclusively by borax and that this upregulation was dose-dependent (Supplementary Fig. S3-a). We have also confirmed NaBC1 upregulation at protein level by In-Cell Western (Supplementary Fig. S3-b).

We next explored whether borax activation of NaBC1 influences the expression of FN-binding integrins. Previous studies have reported the critical role of $\alpha_5\beta_1$[39] and $\alpha_v\beta_3$[40] integrins in osteogenic-adipogenic balance, as well as in regulating gene expression by ion channels, phenomenon only described for potassium channels[41,42]. Figure 2a shows immuno-fluorescence images and staining intensity of $\alpha_5$ and $\alpha_v$ integrins and reveals enhanced integrin levels for PLLA-B2% and PLLA-B5% substrates. We have also detected $\alpha_5$ and $\alpha_v$ integrins at protein level in PLLA-B2% and PLLA-B5% substrates by In-Cell Western (Supplementary Fig. S4-a).

We further analyzed mRNAs of FN-binding integrins from non-transfected MSCs and transfected with esiRNA[NC] and esiRNA[NaBC1], respectively. qPCR analysis showed that active NaBC1 induced mRNA expresion levels of $\alpha_5\beta_1$ and $\alpha_v\beta_3$ integrins (Fig. 2b), effect that was abrogated after NaBC1 silencing. This fact demonstrated that NaBC1 activation is capable of upregulate integrin gene expression as previously reported for other ion channels[41].

Collectively, these findings demonstrate that borax stimulation of NaBC1 leads to enhanced expression of $\alpha_5\beta_1$ and $\alpha_v\beta_3$ integrins, larger spreading area and the greater number of large focal adhesions formed (Fig. 1).

**Active NaBC1 co-localizes with FN-binding integrins and BMPR1A receptors.** We next wanted to study the interplay between NaBC1 and the other membrane receptors involved in the osteogenic pathway. To test whether the simultaneous stimulation of NaBC1 and the $\alpha_5\beta_1$ and $\alpha_v\beta_3$ integrins could follow co-localization, as has previously been described for other ion channels[43], we performed NaBC1/$\alpha_5$-$\alpha_v$ co-localization assays using the DUOLINK® PLA kit system, which shows only the positive signals generated between two different proteins less than 40 nm apart. Similarly, we also investigated NaBC1 co-localization with BMPR1A as the main membrane growth factor receptor that it has been shown to co-localize with integrin $\alpha_v$ to induce osteogenesis, even in absence of its BMP ligands[8] members of the transforming growth factor-β superfamily (TGFβ)[44].

Figure 3a shows the co-localization of NaBC1 with $\alpha_5$ and $\alpha_v$ integrins, as well as co-localization of NaBC1 with BMPR1A. A few fluorescent dots were present in the Glass and PLLA substrates, however, the positive signals in the PLLA-B2% and PLLA-B5% samples increased in a dose-dependent manner, indicating that NaBC1 stimulation promotes the effective co-localization of the receptors, considering that FN-binding integrins are also enhanced by the presence of borax on the FN-coated samples. It is important to note that co-localization of NaBC1/$\alpha_5$ and $\alpha_v$ integrins does not happen at FA sites necessarily. Surprisingly, NaBC1/BMPR1A co-localization occurred even without external addition of BMPs, the BMPR1A natural ligand, as previously reported for osteogenic nanotopo-graphical surfaces[8]. We have also detected an increase of $\alpha_5$ and $\alpha_v$ integrins and BMPR1A at protein level in PLLA-B2% and PLLA-B5% substrates by In-Cell Western (Supplementary Fig. S4). It should be noted that BMPs were not used in this experiment as supplemented-growth factors to induce osteogenesis. These results suggest cooperation between NaBC1/BMPR1A as an osteogenic induction mechanism, in coordination with FN-binding integrins, and in the absence of external growth factors.

**NaBC1 stimulation promotes myosin light chain phosphorylation.** Force transmission after the assembly of focal adhesions is mediated by a Rho-associated kinase (ROCK) pathway, which in turn induces myosin light chain phosphorylation (pMLC) and increases cell contractility[45,46]. To determine whether the observed effect of active NaBC1 on cell adhesion had to do with tension, we analyzed cell cytoskeleton and myosin light chain phosphorylation. Figure 4 shows non-transfected MSCs and transfected with esiRNA[NC] and esiRNA[NaBC1], respectively. PLLA-B2% and PLLA-B5% substrates presented significantly higher levels of pMLC and actin stress fibers than the PLLA and Glass control substrates (Fig. 4a, b and Supplementary Fig. S5-a, b) in non-transfected and esiRNA[NC] transfected cells. However, after transfecting cells with esiRNA[NaBC1], pMLC levels on PLLA-B2% and PLLA-B5% substrates were reduced and resulted similar to PLLA and Glass, demonstrating that cells under NaBC1 stimulation respond by increasing intracellular tension and contractility.

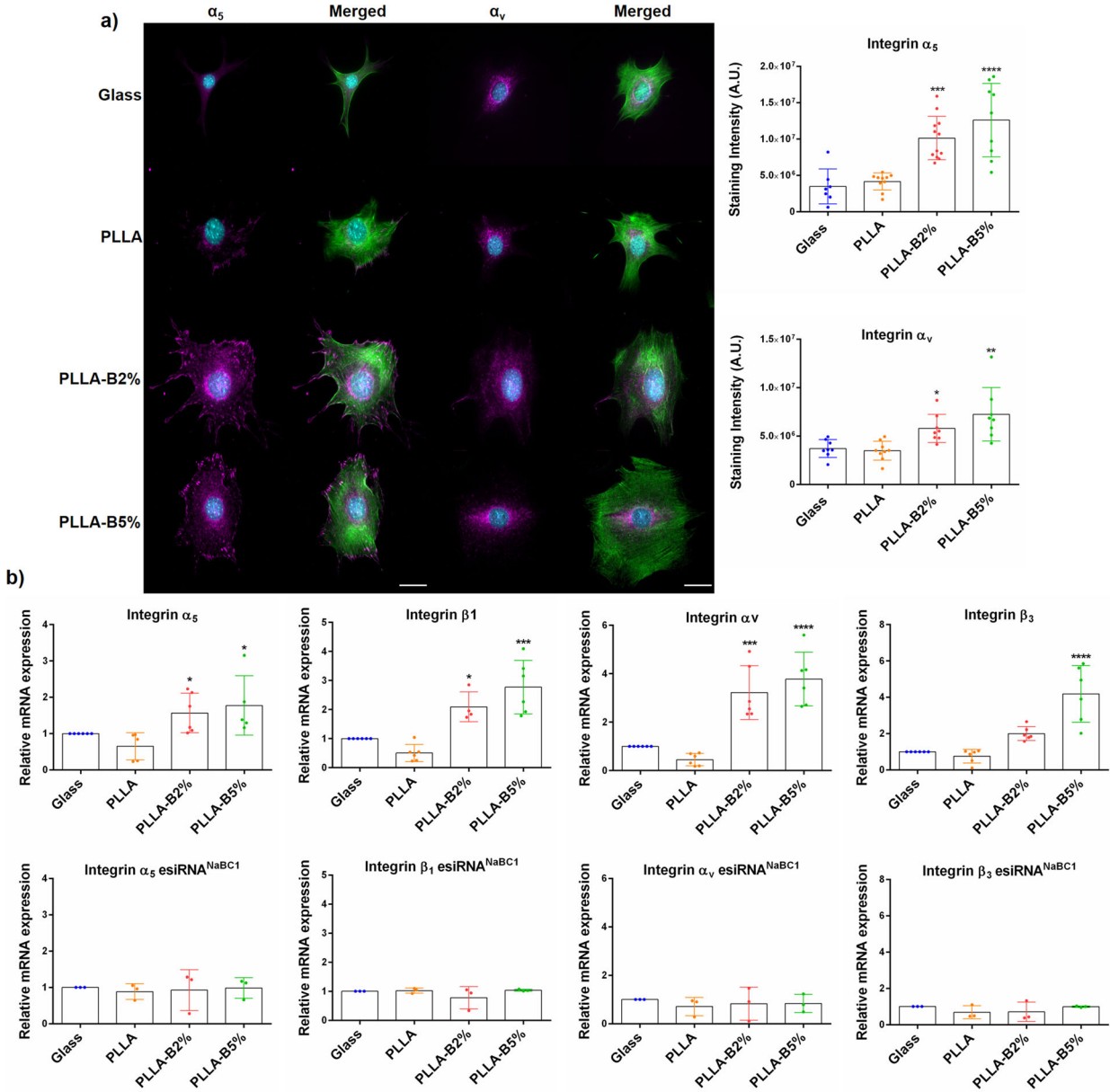

**Fig. 2 NaBC1 induces FN-binding integrins expression. a** Immunofluorescence images of MSCs cells cultured for 3 h onto functionalized substrates (FN-coated), serum-depleted and borax (PLLA-B2%, PLLA-B5%) in culture medium. Images show actin cytoskeleton (green) and $\alpha_5$ or $\alpha_v$ integrins (magenta). Borax induces the expression of FN-binding integrins. Scale bar 25 μm. Right graphs: image analysis quantification of $\alpha_5$ and $\alpha_v$ integrins levels. Twenty images/condition from three different biological replicas were analyzed. The data represented in graphs correspond to $n \geq 7$. **b** qPCR analysis of relative mRNA expression of FN-binding integrins ($\alpha_5\beta_1$ and $\alpha_v\beta_3$) before and after silencing of NaBC1 transporter. Active NaBC1 induces FN-binding integrins expression. $n \geq 3$ different biological replicas. Statistics are shown as mean ± standard deviation. Data were analyzed by an ordinary one-way ANOVA test and corrected for multiple comparisons using Tukey analysis ($P = 0.05$). *$p < 0.05$, **$p < 0.01$, ***$p < 0.001$, ****$p < 0.0001$.

To further investigate the role of active NaBC1 in cell contractility we worked with pharmacological inhibitors that impair contractility. We used Blebbistatin as specific inhibitor of myosin II activity (involved in cytokinesis and cell migration, cortical tension maintenance)[47], and Y-27632 as specific inhibitor of Rho-kinase (disrupt myosin-dependent contractility)[48]. Supplementary Fig. S6 shows immunofluorescence images of actin, vinculin and pMLC (pMyosin) after culturing MSCs with Blebbistatin (Supplementary Fig. S6-a) and Y-27632 (Supplementary Fig. S6-b). As expected, after contractility inhibition MSC morphology was affected in all conditions, however, vinculin and pMLC levels were significantly higher in PLLA-B2% and PLLA-B5% substrates compared with the reduced levels

obtained in PLLA and Glass controls. Note that due to the lack of defined FA formation after using Blebbistatin and Y-27632, we could not perform their quantification. The fact that contractility inhibitors affect to a lesser extent the cells onto PLLA-B2% and PLLA-B5% substrates strongly supports the hypothesis that active NaBC1 reinforce cell adhesion and contractility at early cellular stages.

**Active NaBC1 induced ERK1/2-Smad1-Akt phosphorylation, active YAP-Smad1 nucleus translocation and nuclear stress.** We next wanted to understand the downstream signaling that was triggered by NaBC1 activation. For this, we first explored the

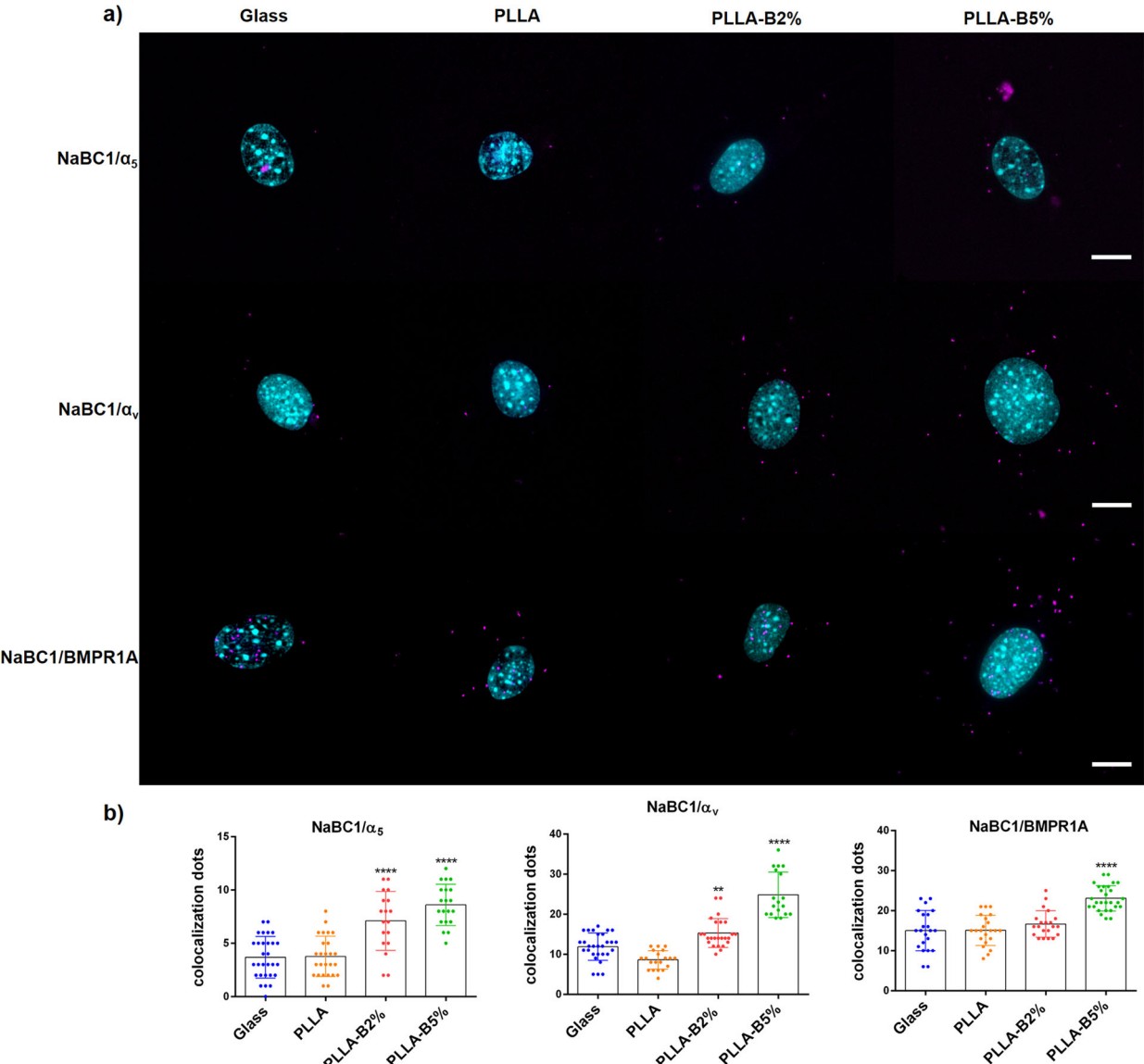

**Fig. 3 NaBC1/α5, αv and NaBC1/BMPR1A co-localization. a** Images of nuclei (cyan) and co-localization dots (magenta) of MSCs cultured for 3 h onto functionalized substrates (FN-coated), serum-depleted and borax (PLLA-B2%, PLLA-B5%) in the culture medium. Scale bar represents 10 μm. **b** Image analysis quantification of the number of co-localization dots of NaBC1/α5, NaBC1/αv and NaBC1/BMPR1A. Co-localization levels increased with borax concentration. Thirty images/condition from three different biological replicas were analyzed. The data represented in graphs correspond to $n \geq 18$. Statistics are shown as mean ± standard deviation. Data were analyzed by an ordinary one-way ANOVA test and corrected for multiple comparisons using Tukey analysis ($P = 0.05$). **$p < 0.01$, ***$p < 0.001$, ****$p < 0.0001$.

phosphorylation of the main effector molecules involved in osteogenic commitment, such as Runt-related transcription factor 2 (Runx2), Extracellular signal-Regulated Kinase 1/2 (ERK1/2), Protein kinase B (Akt) and Small Mothers Against Decapentaplegic 1 (Smad1/5/8). In order to confirm that borax activation of NaBC1 relies on intracellular tension, we also evaluated the active Yes Associated Protein 1 (YAP) presence as another force-induced mechanotransductive marker. In-Cell Western experiments showed that NaBC1 activation significantly induced ERK1/2, Akt, Smad1 phosphorylation (Fig. 5a) and the levels of active cellular YAP, while it had no effect on Runx2.

We further evaluated active Smad1 and YAP intracellular location, as Smad1 tethers with other Smad4 proteins and translocate to the nucleus after phosphorylation, becoming transcriptionally active for the expression of genes involved in osteogenic commitment[49], and YAP is a transcriptional co-

activator acting downstream of the Hippo pathway and one of its substrates is the active nuclear pSmad1[50]. It has been described that high mechanical stress promotes nuclear localization of YAP to drive osteogenesis[51]. Figure 5b shows clearly active Smad1 and YAP nuclear accumulation with a considerable increase only in PLLA-B2% and PLLA-B5% substrates (see magnification images right panel and image analysis quantification).

To quantify the extent to which active NaBC1 affected these regulators of contractility, we quantified nuclear morphology. Addition of borax reduced nuclear circularity while increased nuclear size and the nuclear aspect ratio (AR). Representative images of nuclear morphology for cells on the different substrates show changes in nuclear morphology as borax concentration increases (Fig. 6). These results suggest that activation of NaBC1 might induce intracellular tension and nuclear stress, which alters nuclear morphology. Nuclei are rounded and relaxed for cells on

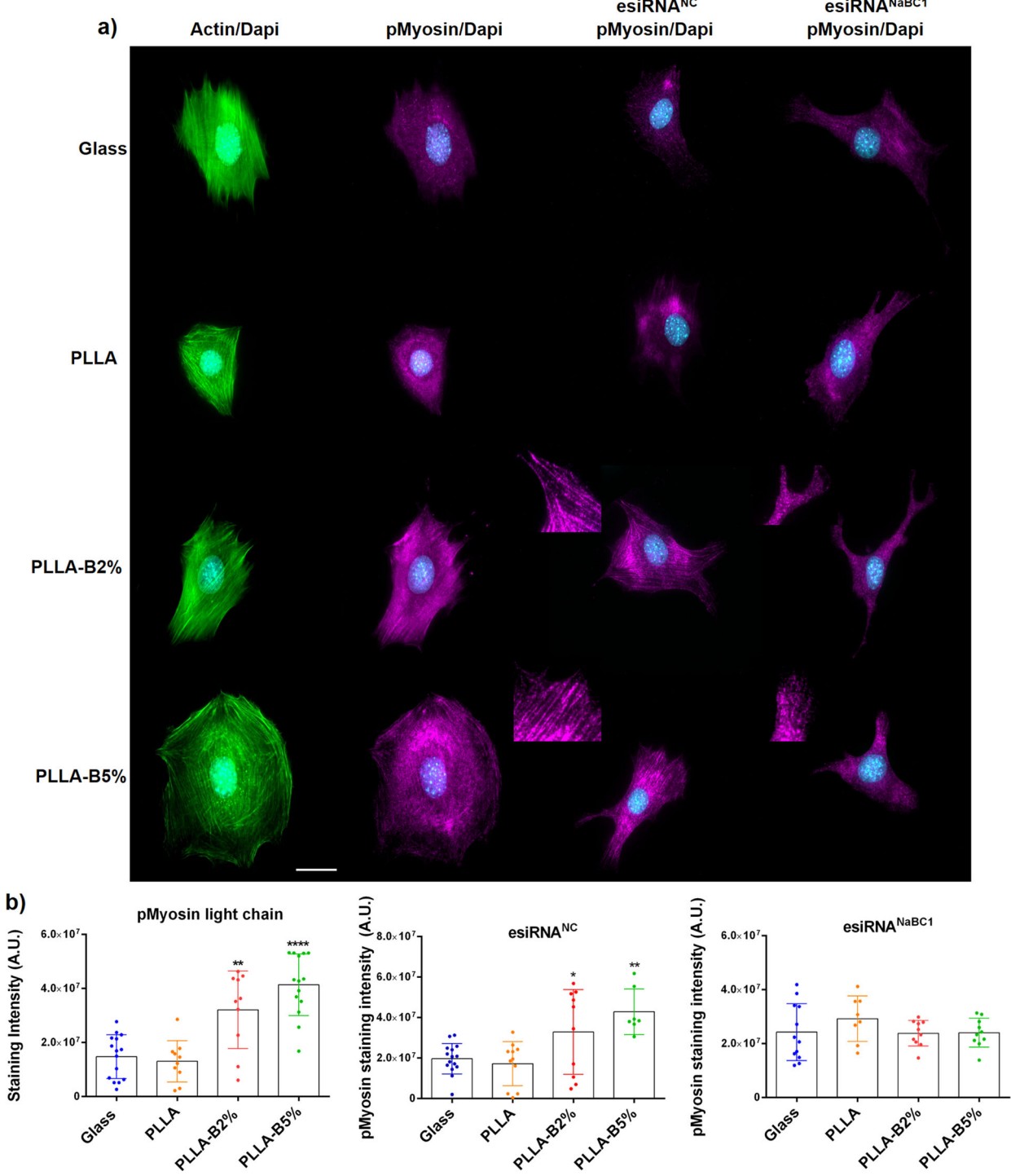

**Fig. 4 Borax effects on myosin light chain phosphorylation and actin stress fiber formation. a** Immunofluorescence images of actin cytoskeleton (green), nuclei (cyan), and pMLC (pMyosin, magenta) as markers of cellular tension and contractility. MSCs were cultured for 3 h onto functionalized substrates (FN-coated), with the presence of serum-depleted and borax (PLLA-B2%, PLLA-B5%) in the culture medium. The same experiment was performed after NaBC1 silencing using esiRNA$^{NC}$ (negative control) and esiRNA$^{NaBC1}$. The PLLA-B2% and PLLA-B5% substrates presented higher levels of actin fibers and pMLC staining that strongly diminished after NaBC1 silencing (see inset magnifications). Scale bar 25 μm. **b** Image analysis quantification before and after NaBC1 silencing of pMLC staining, parameter related to cell contractility. Active NaBC1 induces cell contractility. Fifteen images/condition from three different biological replicas were analyzed. The data represented in graphs correspond to $n \geq 7$. Statistics are shown as mean ± standard deviation. Data were analyzed by an ordinary one-way ANOVA test and corrected for multiple comparisons using Dunnett analysis ($P = 0.05$). *$p < 0.05$, **$p < 0.01$, ****$p < 0.0001$.

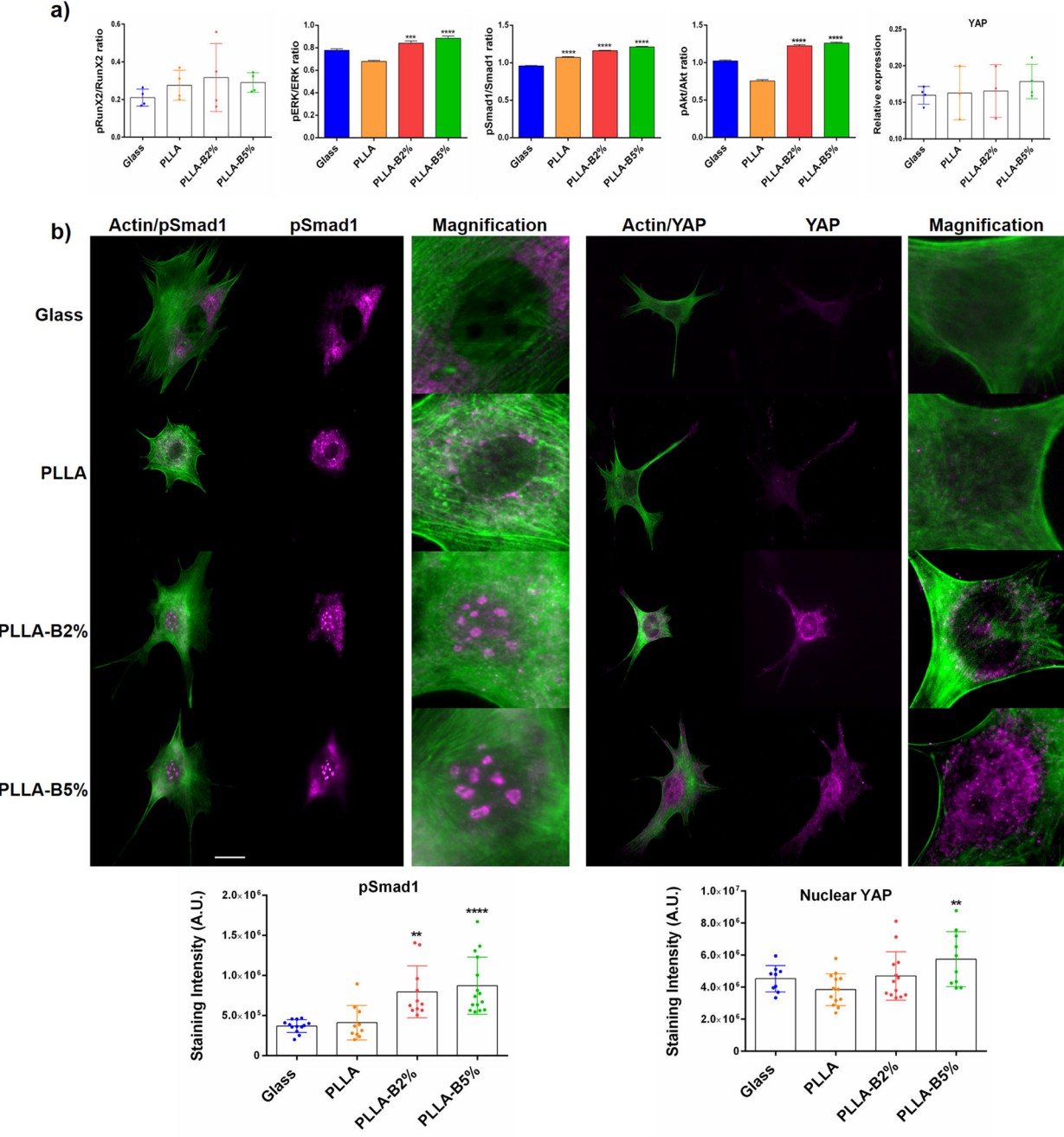

**Fig. 5 Effects of NaBC1 activation in intracellular signaling. a** In-Cell Western assay showing pRunx2/Runx2, pERK/ERK, pAkt/Akt, and pSmad1/Smad1 ratios and active form of total cellular YAP on MSCs cultured onto functionalized substrates (FN-coated) serum-depleted and borax (PLLA-B2%, PLLA-B5%) in the culture medium after 90 min of culture. Simultaneous stimulation of FN-binding integrins and NaBC1 resulted in a significant enhancement of ERK, Smad1, and Akt phosphorylation, as well as active total cellular YAP detection. Graphs showing the ratio between phosphorylated/total proteins are represented as mean ± propagated standard deviation. Statistics for cellular YAP are shown as mean ± standard deviation. $n \geq 3$ different biological replicas. Data were analyzed by an ordinary one-way ANOVA test and corrected for multiple comparisons using Tukey analysis ($P = 0.05$). *$p < 0.05$, **$p < 0.01$, ****$p < 0.0001$. **b** Immunofluorescence images of MSCs cells cultured for 3 h onto functionalized substrates (FN-coated), serum-depleted and borax (PLLA-B2%, PLLA-B5%) in culture medium. Images show actin cytoskeleton (green), pSmad1 and active YAP (magenta). Borax induces active Smad1 and YAP translocation into the cell nucleus (see inset magnifications). Scale bar 25 μm. Image analysis quantification of pSmad1 and active nuclear YAP levels. 15 images/condition from three different biological replicas were analyzed. The data represented in graphs correspond to $n \geq 9$. Statistics are shown as mean ± standard deviation. Data were analyzed by an ordinary one-way ANOVA test and corrected for multiple comparisons using Tukey analysis ($P = 0.05$). **$p < 0.01$, ****$p < 0.0001$.

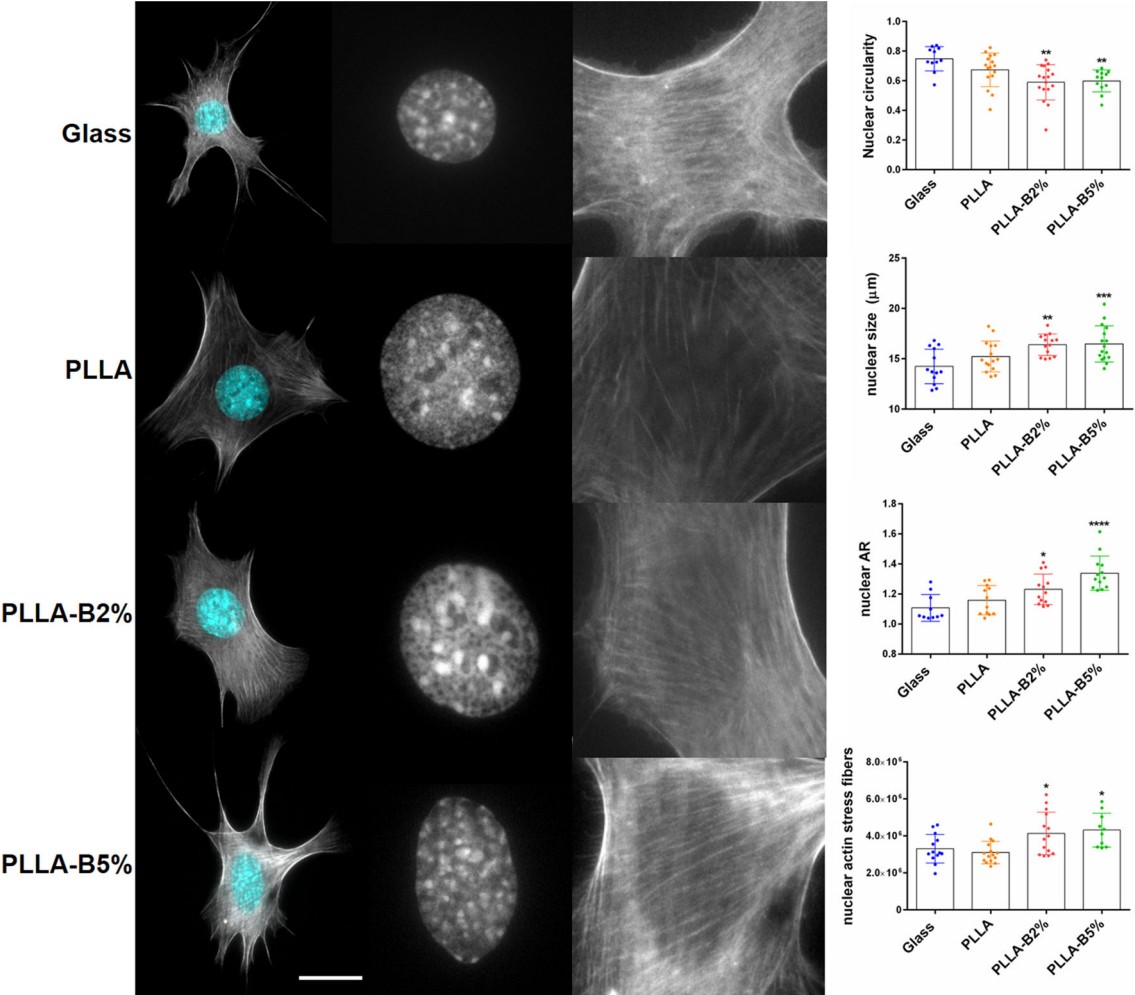

**Fig. 6 Effects of active-NaBC1 in nuclear morphology.** Immunofluorescence images of MSCs cells cultured for 3 h onto functionalized substrates (FN-coated), serum-depleted and borax (PLLA-B2%, PLLA-B5%) in culture medium. Images show actin cytoskeleton (gray) and nuclei (cyan). Borax induces nuclear stress resulting in decreased nuclear circularity, higher nuclear size and aspect ratio (AR), and increased actin stress fibers over the nucleus. Scale bar 20 µm. Image analysis quantification of nuclear circularity, nuclear size, nuclear aspect ratio (AR) and nuclear actin stress fibers. Twenty images/condition from three different biological replicas were analyzed. The data represented in graphs correspond to $n \geq 10$. Statistics are shown as mean ± standard deviation. Data were analyzed by an ordinary one-way ANOVA test and corrected for multiple comparisons using Tukey analysis ($P = 0.05$). *$p < 0.05$, **$p < 0.01$, ***$p < 0.001$, ****$p < 0.0001$.

glass and PLLA substrates and more elongated and stressed on PLLA-B2% and PLLA-B5% substrates. To further confirm this result, we have also quantified the amount of stress fibers over the nucleus. We used the nuclear outline as a mask to crop actin images and the intensity of filamentous actin in this region was quantified. Figure 6 shows that after NaBC1 activation, actin stress fibers over the nucleus increased, supporting the hypothesis that NaBC1 enhances intracellular tension.

**Active NaBC1 stimulates osteogenesis and inhibit adipogenesis in MSCs.** We next evaluated the effect of simultaneous activation of FN-binding integrins and NaBC1 transporter on MSC differentiation. In order to explore the phenotypical and gene expression behavior of MSCs during commitment, we assayed three different experimental conditions: (i) cells grown on basal medium (hereafter Basal), composed of a conventional growing medium containing 10% fetal bovine serum (FBS) without any supplements or growth factors; (ii) cells grown on an osteogenic-specific differentiation medium (hereafter Ob); (iii) cells grown on an adipogenic-specific differentiation medium (hereafter Ad).

For the basal conditions and to promote osteogenesis, cells were seeded at 10,000 cells cm$^{-2}$, whereas cells were seeded at 30,000 cells cm$^{-2}$ to favor adipogenesis[52]. Supplementary Fig. S7-a shows immunofluorescence images of MSCs cultured for 3 days onto FN-coated substrates and (with/without) borax under Basal and Ob conditions for evaluation of Runx2 as an early expressed transcription factor in osteogenenesis[53]. The presence of borax did not increase Runx2 expression either under Basal conditions or under osteogenic stimulation (Ob).

Supplementary Fig. S7b shows the qPCR analysis of specific gene encoding transcription factors involved in the early onset of lineage commitment versus osteogenic (Runx2) and adipogenic (adipocyte peroxisome proliferation-activated receptor-PPARγ2)[54] lineages, under Basal and Ob or Ad conditions. In both cases, borax did not increase either Runx2 or PPARγ2 expression alone, or after the induction of lineage commitment, suggesting that NaBC1 stimulation has no differential effect on the gene expression of these transcription factors at this experimental time point. Previous reports have contrasted the general assumption of the role of Runx2 levels in mediating osteogenesis, showing

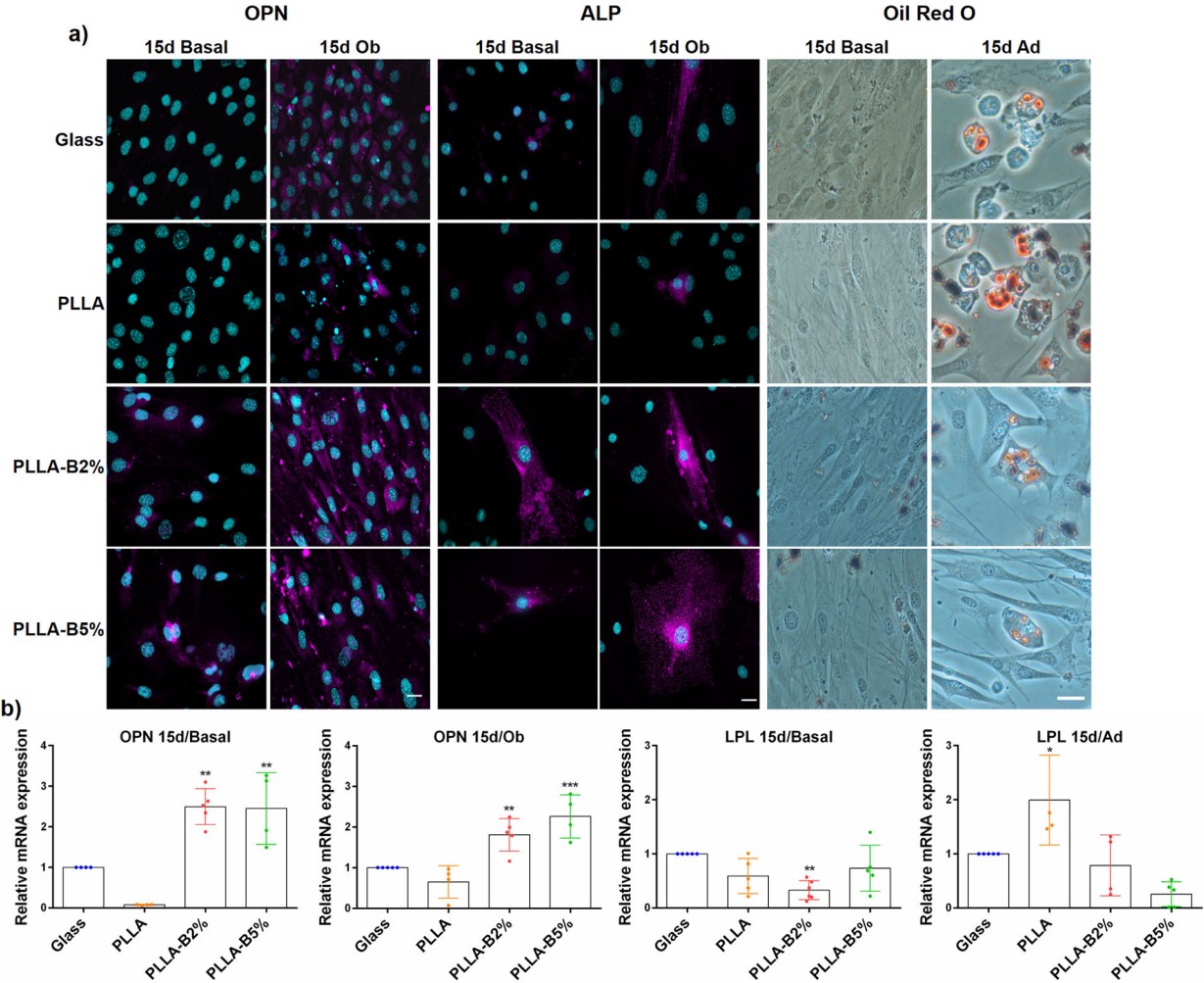

**Fig. 7 Borax effect on MSC differentiation. a** Immunofluorescence images of MSCs cultured for 15 days onto functionalized substrates (FN-coated) and borax (PLLA-B2%, PLLA-B5%) in culture medium, under basal and differentiation conditions (osteogenic), for late osteogenic detection markers (OPN and ALP-magenta, nuclei-cyan). Scale bar 25 μm. Oil red O staining of MSCs cultured for 15 days onto functionalized substrates (FN-coated) and borax (PLLA-B2%, PLLA-B5%) in culture medium, under basal and differentiation conditions (adipogenic), for adipocyte detection showing typical rounded adipocyte shape and accumulation of intracellular fat droplets. Scale bar 25 μm. **b** qPCR analysis of relative mRNA expression of late expressed markers involved in osteogenic (OPN) and adipogenic (LPL) lineage determination. RNA was extracted after 15 days of culture under both basal and differentiation conditions (osteogenic and adipogenic differentiation media). Statistics are shown as mean ± standard deviation. $n \geq 3$ different biological replicas. Data were analyzed by an ordinary one-way ANOVA test and corrected for multiple comparisons using Tukey analysis ($P = 0.05$). *$p < 0.05$, **$p < 0.01$, ***$p < 0.001$.

Runx2 constitutive expression and no correlation between Runx2 mRNA/protein levels and the osteoblast phenotype[53].

Figure 7a shows representative images of late expressed markers involved in osteogenic (osteopontin-OPN and alkaline phosphatase-ALP)[53] and adipogenic (adipocyte formation) differentiation after 15 days of culture[55]. MSCs were cultured for 15 days onto FN-functionalized substrates and borax under Basal and Ob or Ad conditions. NaBC1 stimulation induced higher OPN and ALP staining levels in an osteogenic defined medium than in PLLA and Glass control conditions. Surprisingly, the capacity of the sole presence of borax to promote osteogenic lineage commitment was particularly strong, as revealed by the higher levels of OPN and ALP under Basal conditions (Fig. 7a). We have obtained similar results after histological analysis of Alizarin red, ALP and Von Kossa staining (Supplementary Fig. S7-c) and immunofluorescence detection of Collagen I (Col I), Integrin Binding Sialoprotein (IBSP) and Osteocalcin (OCN) (Supplementary Fig. S7-d). Conversely, MSCs cultured for 15 days onto FN-functionalized

substrates and borax under Ad conditions gave the opposite results. Adipocyte formation after Oil Red O staining was evident only in the Glass and PLLA substrates, showing the clear formation of rounded cells with lipid vacuoles containing fat droplets[55]. PLLA-B2% and PLLA-B5% substrates inhibited adipocyte formation, and kept the polygonal morphology typical of osteoblasts with a minimal amount of red-stained fat droplets (Fig. 7a).

We also investigated the gene expression of the late expressed markers involved in osteogenic (osteopontin-OPN) and adipogenic (Lipoprotein lipase-LPL)[53,55] differentiation by qPCR after 15 days of culture. Figure 7b shows that NaBC1 and FN-binding integrin co-stimulation strongly induced OPN expression under Basal and osteogenic differentiation conditions, whereas they extremely reduced LPL levels, results that are in concordance with the immunofluorescence images shown in Fig. 7a and those obtained for focal adhesion formation.

These results demonstrate that the simultaneous stimulation of NaBC1 and FN-binding integrins promotes osteogenesis (in basal

conditions) and strongly inhibits adipogenesis, even in an adipogenic induction medium.

## Discussion

Understanding the role of ECM and its ability to interact with membrane receptors to mimic natural cellular niches is critical for the success of tissue engineering approaches. Diverse strategies have been used to control MSC growth and differentiation, including: modulation of the physical properties of materials in terms of chemistry[4], stiffness[5,6] and topography[7,8], controlling spatio-temporal growth factor delivery/exposure to mimic cell receptor-growth factor crosstalk and activate the signaling pathways involved in MSC fate[9], or even combining modifications of their material properties by soluble growth factor approaches[56]. As has been broadly described in the literature, cell mechanotransduction mechanisms are the key determinants of MSC physiology[5,11,56,57]. Even though several reports have described ion channels as mechanosensors[15,16], whether their contribution to MSC behavior is caused by force transmission or is simply due to the role of ion homeostasis in intracellular mechanisms is still not clear, and the studies published to date focus mainly on $Ca^{2+}$, $K^+$, and $Cl^-$ channels[17].

This study demonstrates that the borate NaBC1 transporter, acts by co-localizing with BMPR1A and activating the BMP-dependent downstream pathways in the absence of any external BMP supplementation, driving the MSCs to osteogenic commitment.

As BMP2 and 4 are known to be constitutively synthesized by C3H10T1/2 in a given osteogenic medium[59], the role of BMPs in BMPR1A activation in long term cultures for MSC differentiation under certain conditions cannot be ruled out. However, as in our co-localization and In-Cell Western experiments we used serum-depleted basal media only, without any osteogenic induction, we can propose a not described crosstalk mechanism involving active NaBC1 and BMPR1A in the absence of external BMPs.

Recent reports have described the effects of boron in the induction of osteogenesis[23–25] and inhibition of adipogenesis[26,27]. Even though they do describe the possible metabolic pathways that explain the observed effects (mainly on adipogenic inhibition), they do not describe the mechanism used to transport boron and the intracellular signaling mechanisms induced after activation of NaBC1 transporter.

Our initial hypothesis was that the simultaneous activation of membrane receptors enhances intracellular signaling, as previously described for cooperation mechanisms between integrins and growth factor receptors[60]. To test this hypothesis, we employed FN-coated substrates for $\alpha_5\beta_1$ and $\alpha_v\beta_3$, and borax for NaBC1 stimulation, respectively. In this study, we used borax solution ($Na_2B_4O_7$), so that the B uptake by the cells would be guaranteed via NaBC1, since this is an obligated $Na^+$-coupled borate co-transporter[18]. We have shown that the simultaneous activation of NaBC1 and FN-binding integrins enhances cell adhesion in terms of greater cell spreading and the formation of mature focal adhesions (Fig. 1 and Supplementary Fig. S2). The transient depletion of NaBC1 function, after transfecting cells with esiRNA$^{NaBC1}$, clearly points out the synergistic effect of NaBC1 transporter in combination with FN-binding integrins, and strongly suggest that NaBC1 activation regulates cell adhesion. This enhanced adhesion (in FA size and number) determines the MSCs differentiation towards osteogenic commitment, while it inhibits adipogenesis, as expected, being the osteogenic-adipogenic balance is mutually exclusive (Fig. 7).

Our results fit in nicely with previous reports describing the importance of adhesion size, normally associated with cell motility (small adhesions, low intracellular tension derives into adipocytes) or high stability and cytoskeletal tension (long and mature adhesions, high intracellular tension derives into osteoblasts)[57]. Despite our findings on adhesion, we did not find any significant differences in the pFAK/FAK ratio (Supplementary Fig. S2-e). As FAK is one of the main downstream effectors after integrin activation[7], this result was unexpected, and suggests that simultaneous stimulation of NaBC1 and FN-binding integrins involves other metabolic pathways rather than FAK phosphorylation, as we previously reported[21]. To verify this finding we explored the phosphorylation of the myosin light chain (pMLC) involved in cell contractility and acting downstream of the FAK. The results show that NaBC1 activation induced elevated levels of pMLC and actin stress fibers (Fig. 4 and Supplementary Fig. S5), effects reverted after NaBC1 silencing, specifying NaBC1 activation in the promotion of cell adhesion and contractility. The fact that after using contractility inhibitors pMLC and vinculin levels remained elevated when simultaneous stimulation of NaBC1 and FN-binding integrins occurred (Supplementary Fig. S6), strongly suggest that the induction of cytoskeleton tension is immediate in PLLA-B2% and PLLA-B5% substrates. Cooperation between NaBC1/FN-binding integrins can produce a more robust strengthening response.

Our results show that the simultaneous activation of NaBC1 and FN-binding integrins causes an adhesion-primed state of the MSCs related to intracellular tension, caused by the increased FA, pMLC and stress actin fibers, and elevated $\alpha_5\beta_1/\alpha_v\beta_3$ integrins and NaBC1 expression (Fig. 2, Supplementary Figs. S3 and S4). NaBC1/$\alpha_5\beta_1/\alpha_v\beta_3$ and NaBC1/BMPR1A clustering and co-localization (Fig. 3) may act as a cooperation switch affecting the downstream effectors of TGFβ and Hippo pathways, Smad1 and YAP, respectively, which became active and translocate to the nucleus to exert their transcriptional function[49,50] (Fig. 5). Indeed, the analysis of nuclear morphology demonstrate that this cooperation mechanism induces more elongated and stressed nuclei, as well as increased actin stress fibers over the nucleus, supporting the hypothesis that active NaBC1 increases intracellular tension (Fig. 6). However, further studies will be needed to establish the precise role of NaBC1 as a mechanosensor transporter, including their response on surfaces of controlled elastic properties.

Some studies have described the role of FN-binding integrins in osteogenic-adipogenic balance[39,60,61], the role of BMPR1A driving osteogenesis after BMP ligand activation[62], and describing YAP as the key determinant of cell mechanics that controls focal adhesion assembly[63]. Even though Runx2 is a key target for active Smad1 and has been proposed as the main mediator of downstream BMP actions, several authors report that some BMP effects are Runx2-independent in bone formation[58]. We did not find significantly high levels of this osteogenic regulator gene in our experimental system, but instead we did find higher OPN levels expression (even in Basal conditions). OPN-integrin interaction has been described as critical for MSCs osteogenic differentiation[40], and thus higher OPN expression, obtained by the sole addition of borax (Fig. 7) supports the crosstalk mechanism between active NaBC1/FN-binding integrins/BMPR1A. Another pSmad1 target is Hoxc-8, a transcriptional repressor that liberates the transcription of the OPN gene[64] after pSmad1 binding and agrees with our hypothesis.

We have therefore demonstrated a novel function for the NaBC1 transporter in MSCs. To date, the synergistic interactions between ion transporters and other membrane receptors have not been exploited to engineer material systems for biomedical applications. We here propose a simple approach for driving osteogenesis (and in turn inhibiting adipogenesis) through a mechanism that involves the simultaneous stimulation of NaBC1, BMPR1A and $\alpha_5\beta_1/\alpha_v\beta_3$ integrins to enhance intracellular

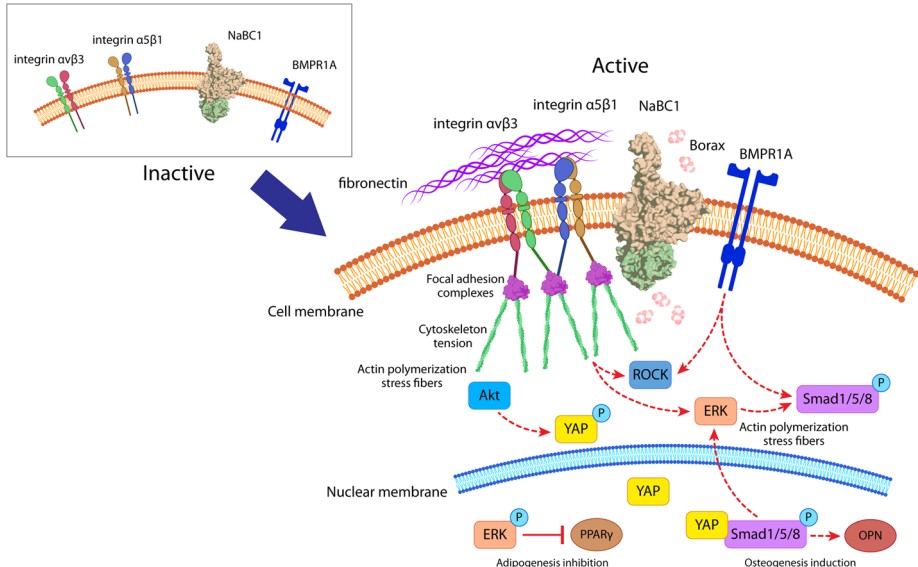

**Fig. 8 Molecular control of spatiotemporal cues for synergistic intracellular signaling.** Description of molecular events occurring after simultaneous NaBC1/$\alpha_5\beta_1$/$\alpha_v\beta_3$ and NaBC1/BMPR1A activation. FN stimulate FN-binding integrins while borax stimulates the NaBC1 transporter. NaBC1 co-localizes with FN-binding integrins and BMPR1A, thus cooperating and stimulating downstream pathways controlling MSCs fate. Note that this cooperation does not necessarily implies a physical interaction. Canonical BMPR1A activation concludes in Smad1 phosphorylation, inducing osteogenesis via OPN expression. After activation, both (NaBC1 and FN-binding integrins) receptors induce the formation of large focal adhesions, resulting in high intracellular tension that derive into nuclear stress and YAP translocation to the nucleus inducing osteogenesis. High mechanical tension activates ERK, which inhibits adipogenesis after phosphorylation.

signaling. As the co-localization experiments show that these receptors may cooperate between them and activates the translocation of Smad1 and YAP into the nucleus, we thus propose NaBC1 as a novel transporter enhancing intracellular signaling besides controlling borate homeostasis. Our results indicate an adhesion-primed state related to intracellular tension involving the diverse crosstalk mechanisms described in Fig. 8. These findings will open up new ways for engineering biomaterials to be employed in biomedical applications avoiding the use of high doses of soluble growth factors.

## Conclusions

This work proposes an original way to control MSC fate using the interplay between specific cell membrane receptors. Our data show that the simultaneous stimulation of NaBC1 and FN-binding integrins generates an adhesion-primed state (mature FA, cell spreading), stimulating the mechanical determinants that affect cell fate through actomyosin forces (pMLC, actin stress fibers, nuclear stress), and the nuclear translocation of transcription factors (pSmad, YAP). This stimulation enhances osteogenesis and inhibits adipogenesis in the absence of other soluble chemicals or growth factors. We describe an innovative mechanism involving the cooperation and crosstalk of active NaBC1/$\alpha_5\beta_1$/$\alpha_v\beta_3$ integrins and NaBC1/BMPR1A and the intracellular pathways involved, with a novel function for NaBC1 as more than a merely borate transporter.

## Methods

**Material substrates**. Cleaned glass cover slips were used as control substrates. PLLA (Cargill Dow) was dissolved at a concentration of 2% (w/v) in chloroform. Sodium Tetraborate Decahydrate Borax 10 Mol (hereafter borax) ($Na_2B_4O_7\cdot10H_2O$, Borax España S.A) was dissolved in the culture medium in all the experiments. Two different borax concentrations were used, 0.59 mM and 1.47 mM, respectively, in the PLLA-B2% and PLLA-B5% samples. PLLA 2% (w/v) solution was used to prepare thin films by spin-coating on cleaned glass cover slips for 30 s at 3000 rpm. Samples were dried at 60 °C in vacuo for 4 h.

All the substrates were functionalized with human plasma fibronectin (FN, Sigma-Aldrich). After sterilizing with UV for 30 min, the substrates were coated

with 20 µg mL$^{-1}$ FN solution in Dulbecco's Phosphate Saline Buffer (DPBS) for 1 h at room temperature.

**Cytotoxicity assay**. MTS quantitative assay (The CellTiter 96 Aqueous One Solution Cell Proliferation Assay, Promega) was performed to assess cyto-compatibility of borax with MSCs and to establish the maximum working concentrations to use with this particular cell line. 20,000 cells cm$^{-2}$ were seeded onto a p-24 multi-well plate and metabolic activity was measured after 24 h of incubation of cells with different quantities of borax in media (0.2, 0.6, 3, 6, 10, and 20 mg mL$^{-1}$). Cells were then incubated for 3 h with MTS (tetrazolium salt) at 37 °C and the formation of formazan was followed by measuring absorbance at 490 nm. All measurements were performed in triplicate.

**MSCs culture**. Murine embryonic mesenchymal stem cells (MSCs) C3H10T1/2 (RIKEN Cell Bank, Japan) were cultured in Dulbecco's Modified Eagle Medium (DMEM) with high glucose content, 10% fetal bovine serum (FBS) and 1% anti-biotics (penicillin/streptomycin) at 37 °C in a humidified atmosphere of 5% CO$_2$ (conventional growth medium and basal medium composition (hereinafter Basal). Cells were subcultured once a week before reaching confluence.

**MSC transfection**. MSCs were seeded at 60,000 cells cm$^{-2}$ in Dulbecco's Modified Eagle Medium (DMEM) with high glucose content, 10% fetal bovine serum (FBS) and 1% antibiotics (penicillin/streptomycin) at 37 °C in a humidified atmosphere of 5% CO$_2$. After 24 h, cells were washed with Opti-MEM reduced serum medium (ThermoFisher) and transfected using pre-designed MISSION esiRNA (Sigma-Aldrich) against mouse NaBC1 transporter in X-tremeGENE siRNA Transfection Reagent (Roche), following manufacturer's instructions. MISSION siRNA Fluor-escent Universal Negative Control 1, Cyanine 3 (NC, Sigma-Aldrich) was used as a control of transfection efficiency.

**Cell adhesion experiments**. For cell adhesion experiments, MSCs or esiRNA transfected MSCs were plated on glass covers and PLLA substrates functionalized with FN at a density of 5000 cells cm$^{-2}$. Cells were cultured in DMEM with high glucose content, 1% antibiotics and in absence of serum. Samples were also supplemented with 0.59 mM or 1.47 mM from borax solution as required. After 3 h at 37 °C cells were fixed and immunostained.

**Cell differentiation experiments**. For differentiation experiments, MSCs were plated on glass covers and PLLA substrates functionalized with FN at a density of 20,000 cells cm$^{-2}$ for osteogenic and basal conditions and 30,000 cells cm$^{-2}$ for adipogenic differentiation. Cells were cultured for 48 h in DMEM growth media until they reached 70–80% confluence. After confluence, differentiation was induced with osteogenic media (hereinafter Ob) (DMEM growth medium supplemented with

Ascorbic Acid 50 μg mL$^{-1}$, Glycerophosphate 10 mM and Dexamethasone 0.1 μM) or adipogenic media (hereinafter Ad) (DMEM growth medium supplemented with 3-isobutyl-1-methyl-xanthine (IBMX) 0.5 mM, indomethacin 60 μM and Hydrocortisone 0.5 μM). Samples were also supplemented with 0.59 mM or 1.47 mM of borax solution in every change of medium as required. Media was changed every 3 days until end-point assay. All differentiation experiments were finished after 3 days for analysis of early transcription factors (Runt-related transcription factor 2—Runx2—and Peroxisome proliferator-activated receptor gamma-PPAR-γ) or 15 days for analysis of osteogenic/adipogenic markers (Osteopontin–OPN, Osteocalcin–OCN, Integrin Binding Sialoprotein–IBSP, Alkaline Phosphatase–ALP, Lipoprotein lipase–LPL or adipocyte formation).

**Immunohistochemistry assays and staining.** After culture, cells were fixed in a 4% formalin solution (Sigma-Aldrich) at 4 °C for 30 min. Samples were then rinsed with DPBS and permeabilized with DPBS/0.5% Triton X-100 at room temperature for 5 min. Samples were then incubated with primary antibodies in blocking buffer DPBS/2% BSA at 37 °C for 1 h or overnight at 4 °C. The samples were then rinsed twice in DPBS/0.1% Triton X-100 and incubated with the secondary antibody and/or BODIPY FL phallacidin (Invitrogen, 1:100) at room temperature for 1 h. Finally, samples were washed twice in DPBS/0.1% Triton X-100 before mounting with Vectashield containing DAPI (Vector Laboratories) and observed under an epifluorescence microscope (Nikon Eclipse 80i).

For cell adhesion studies monoclonal antibodies against vinculin-FA detection (Sigma-Aldrich, 1:400), pMLC-intracellular tension (Cell Signaling, 1:200), integrin α$_5$ (abcam, 1:500), integrin α$_v$ (abcam, 1:500), pSmad1 (Cell Signaling, 1:200), active YAP (abcam, 1:500) were used. Cy$^3$ conjugated (Jackson Immunoresearch, 1:200) or Alexa Fluor 555 (ThermoFisher, 1:700) were used as secondary antibodies.

Several specific markers were used to evaluate differentiation. Osteogenic differentiation was assessed by Runx2 (abcam, 1:100), Osteopontin (Santa cruz Biotechnology, 1:100), Osteocalcin (abcam, 1:200), Integrin Binding Sialoprotein (Santa Cruz Biotechnology, 1:200), Collagen I (abcam, 1:200) as primary antibodies. Cy$^3$ (Jackson Immunoresearch, 1:200) and Alexa Fluor 488 or 555 (Invitrogen, 1:700) were used as secondary antibodies.

**Osteogenic differentiation was assessed also by histological staining.** Alkaline phosphatase activity (ALP) was determined staining samples with naphtol AS-MX/Fast Red TR (Sigma-Aldrich) for 30 min at room temperature. After incubation, samples were washed and nuclei were labeled with Hoechst. These samples can be visualized either histologically or by fluorescence microscopy.

Von Kossa and Alizarin red staining were performed following the PROMOCELL procedure.

Adipogenic differentiation was detected by observation of lipid levels that were qualitatively assessed by a standard Oil Red O staining protocol. Briefly, cells were washed in DPBS. Immediately before use, 30 mL of a stock solution of Oil Red O (3 mg mL$^{-1}$ in 99% isopropanol) was mixed with 20 mL diH$_2$O, filtered and applied for 10–15 min at room temperature to cells pre-equilibrated with 60% isopropanol.

**Gene expression.** Total RNA was extracted from MSCs cultured for 3 or 15 days under different experimental conditions using RNeasy Micro Kit (Qiagen). RNA quantity and integrity was measured with a NanoDrop 1000 (ThermoScientific). Then 500 ng of RNA were reverse transcribed using the Superscript III reverse transcriptase (Invitrogen) and oligo dT primer (Invitrogen). Real-time qPCR was performed using Sybr select master mix and 7500 Real Time PCR system from Applied Biosystems. The reactions were run at least in triplicate for both technical and biological replicas. The primers used for amplification were designed from sequences found in the GenBank database and included:

*Runx2* (NM_001146038.1, Forward: 5′-TGA GAG TAG GTG TCC CGC CT-3′, Reverse: 5′-TGT GGA TTA AAA GGA CTT GGT GC-3′) and *Osteopontin* (NM_001204201.1, Forward: 5′-TTT GCC TGT TTG GCA TTG C-3′, Reverse: 5′-TGG GTG CAG GCT GTA AAG CT-3′) for osteogenic differentiation. *PPARγ2* (NM_001127330.1, Forward: 5′-AGC AAA GAG GTG GCC ATC C-3′, Reverse: 5′-CTT GCA CGG CTT CTA CG-3′) and *LPL* (NM_008509.2, Forward: 5′-TGC CCT AAG GAC CCC TGA A-3′, Reverse: 5′-CAG TTA GAC ACA GAG TCT GC-3′) were used for adipogenic differentiation.

*NaBC1* (NM_001081162.1, Forward: 5′-GAG GTT CGC TTT GTC ATC CTG G-3′, Reverse: 5′-TTC CTC TGT GCG AGT CTT CAG G-3 were used for boron transporter amplification and *GAPDH* (NM_008084.2, Forward: 5′-GTG TGA ACG GAT TTG GCC GT-3′, Reverse: 5′-TTG ATG TTA GTG GGG TCT CG-3′) were used as a housekeeping gene. *Integrin α$_5$* (NM_010577.3, Forward: 5′-GGA CGG AGT CAG TGT GCT G-3′, Reverse: 5′-GAA TCC GGG AGC CTT TGC TG-3′), *Integrin β$_1$* (NM_010578, Forward: 5′-CAT CCC AAT TGT AGC AGG CG-3′, Reverse: 5′-CGT GTC CCA CTT GGC ATT CAT-3′), *Integrin α$_v$* (NM_008402.2, Forward: 5′-CAC CAG CAG TCA GAG ATG GA-3′, Reverse: 5′-GAA CAA TAG GCC CAA CGT TC-3′), *Integrin β$_3$* (NM_016780.2, Forward: GGA ACG GGA CTT TTG AGT GT-3′, Reverse: 5′-ATG GCA GAC ACA CTG GCC AC-3′) and *β-actin* (NM_007393.3, Forward: 5′-TTC TAC AAT GAG CTG CGT GTG-3′, Reverse: 5′-GGG GTG TTG AAG GTC TCA AA-3′) were used as a housekeeping gene.

The fractional cycle number at which fluorescence passed the threshold (Ct values) was used for quantification by the comparative Ct method. Sample values were normalized to the threshold value of housekeeping gene *GAPDH* or *β-actin*: $\Delta C_T = C_T(experiments) - C_T(\beta-actin)$. The Ct value of the control (B0% substrate) was used as a reference. $\Delta\Delta C_T = \Delta C_T(experiments) - \Delta C_T(control)$. mRNA expression was calculated by the following equation: fold change $= 2^{-\Delta\Delta C_T}$.

**Co-localization experiments.** Co-localization of NaBC1/BMPR, NaBC1/α$_5$ and α$_v$ experiments were performed using DUOLINK® PLA system (Sigma-Aldrich) and following the manufacturer's instructions. Specific primary antibodies used were: anti-NaBC1 (abcam, 1:200), anti-BMPR1A (abcam, 5 μg mL$^{-1}$), anti-integrin α$_5$ (abcam, 1:500) and anti-integrin α$_v$ (abcam, 1:500). For image quantification of co-localization fluorescent dots, at least 30 individual cells were imaged for each condition under an epifluorescence microscope (Nikon Eclipse 80i).

**In-cell western.** For evaluation of pMLC, ERK, Runx2, FAK, Akt and Smad1 phosphorylation, as well as active YAP, NaBC1, integrin α$_5$ and α$_v$ we used In-Cell Western quantification. MSCs (10,000 cells cm$^{-2}$) were seeded onto FN-coated substrates during 1.5 h at 37 °C and 5% CO$_2$. Cells were then fixed using fixative buffer (10 mL formaldehyde, 90 ml PBS, 2 g sucrose) at 37 °C for 15 min and then permeabilized in cold methanol at 40 °C for 5 min. Cells were then blocked in 0.5% blocking buffer (non-fat dry milk powder in 0.1% PBST buffer) at RT for 2 h followed by 3 washes of 10 min with 0.1% PBST. Cells were then incubated with primary antibodies: pMLC (Cell Signaling, 1:200), Runx2 and pRunx2 (Stratech, 1:100), ERK and pERK (Cell Signaling, 1:200), Smad1 and pSmad1 (Cell Signaling, 1:200), FAK and pFAK (Millipore, 1:200), Akt and pAkt (ThermoFisher, 1:500), active YAP (abcam, 1:500), NaBC1 (abcam, 1:200), BMPR1A (abcam, 5 μg mL$^{-1}$), integrin α$_5$ (abcam, 1:500) and anti-integrin α$_v$ (abcam, 1:500) diluted in blocking buffer at 4 °C overnight. After 3 washes of 10 min with 0.1% PBST buffer, cells were incubated with 1:800 diluted infrared-labeled secondary antibody IRDye 800CW (LI-COR) and 1:500 diluted CellTag 700 Stain (LI-COR) at RT for 1 h, followed by 5 washes of 10 min with 0.1% PBST. Samples were then dried overnight at room temperature. Infrared signal was detected using an Odyssey infrared imaging system.

**Image analysis.** To analyze focal adhesions, vinculin images were segmented by ImageJ, using Trainable Weka Segmentation plugin to create a binary mask. After segmentation, focal adhesion size and number were determined using different commands of the same software. Values of focal adhesion size frequency were represented using GraphPad Prism 6.0. using a bin width of 0.2 μm. Cell and nuclear morphology were analyzed by calculation of different parameters using ImageJ software. Staining intensity of immunofluorescence images were quantified by ImageJ software.

**Statistics and reproducibility.** For statistical analysis, the data were analyzed for normality using the D'Agostino and Pearson omnibus normality test with an alpha of 0.05. When the normality test was passed, an ordinary one-way ANOVA test with a Tukey's, Sidak's or Dunnett's post-hoc analysis ($p = 0.05$) was used to compare the means of the columns against the control column. When the normality test was not passed, a non-parametric test with a post-hoc Dunn analysis ($p = 0.05$) was used to compare the means of each column against the control column. Data is represented as mean ± standard deviation. Sample size for each statistical analysis is indicated in the corresponding figure legends. GraphPad Prism 6 XML software has been used for statistical analysis.

**Reporting summary.** Further information on research design is available in the Nature Research Reporting Summary linked to this article.

## Data availability

The datasets generated during and/or analyzed during this study are available in the University of Glasgow Repository, http://researchdata.gla.ac.uk/[65].

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

## Acknowledgements

P.R. acknowledges support from the Spanish Ministry of Science, Innovation and Universities (RTI2018-096794), and Fondo Europeo de Desarrollo Regional (FEDER). CIBER-BBN is an initiative funded by the VI National R&D&I Plan 2008-2011, Iniciativa Ingenio 2010, Consolider Program, CIBER Actions and financed by the Instituto de Salud Carlos III with assistance from the European Regional Development Fund. M.S.S. acknowledges support from the UK Engineering and Physical Sciences Research Council (EPSRC-EP/P001114/1).

## Author contributions

Conceptualization: P.R. Methodology: P.R., A.R.-N., L.S.P. Investigation: P.R. and M.S.-S. Writing: P.R. Review and editing P.R. and M.S.-S. Supervision: P.R.

## Competing interests

The authors declare no competing interests.
