## [Peer Review File · Communications Biology]

Reviewers' comments:

Reviewer #1 (Remarks to the Author):

The paper by Dr. Rico et al. shows that, in mesenchymal stem cells, stimulation of the NaBC1 transporter can enhance integrin-dependent cell adhesion and contractility. Moreover, stimulation promotes osteogenesis and inhibits adipogenesis. Osteogenic differentiation is found to depend on activation of the BMP pathway. The process involves co-localization of NaBC1 with integrins and BMPRI1A.

Overall, the data are interesting. Elucidating the complex mechanisms linking membrane transporters to other membrane proteins in the regulation of cell adhesion, motility and differentiation, in the context of the cell's microenvironment, is an important and timely topic.

There are however several aspects of the authors' interpretation that are not fully convincing, especially concerning the first steps of the cellular mechanism (mechanosensitivity and protein-protein interaction). Demonstrating these issues would require substantial further experimentation. Alternatively, the text should be modified to remain consistent with the experimental evidence as well as clarify several unclear points.

MAJOR ISSUES.

1) The authors repeatedly refer to NaBC1 as an ion channel. However, in the presence of borate, NaBC1 actually behaves as a Na⁺-dependent borate transporter (e.g. Park et al. Mol Cell 2004). This is stated by the authors themselves (e.g. on p. 4), who also add (p. 15) that the natural ligand is borax, for which no passive diffusion is known to occur. Therefore, NaBC1 normally performs a secondary active transport, which is conceptually different from a passive flow through a conduction pore (which is typical of ion channels). Is there any evidence that these transporters have channel-like states also in the presence of borate (and not only in the absence of the latter)? If so, this should be thoroughly discussed in the paper. Otherwise, NaBC1 should not be referred to as an ion channel.

2) If NaBC1 is not an ion channel, it cannot be a mechanosensitive channel (Abstract and elsewhere), but could nonetheless be a mechanosensitive transporter. However, nowhere in the paper demonstration is provided that mechanical tension directly activates the transport mechanism. In the case of stretch-activated ion channels, this could be demonstrated by studying the membrane currents modulated by mechanical stimulation. Unfortunately, this procedure can be rarely applied to electrogenic transporters, whose currents have very small amplitudes. Hence, to investigate mechanotransduction in NaBC1, one should attempt to produce a suitable ad hoc experimental model (e.g. overexpressing the transporter, etc.). Another option would be to exploit the channel-like behaviour observed in the absence of borate (although this would somewhat decrease the physiological significance).

Nevertheless, I think the hypothesis that NaBC1 is a mechanosensor is not necessary for the authors' purposes. What the data show is that activation of the transporter cooperates with other membrane receptors to regulate adhesion and the contractility machinery. It is not necessary to assume an early NaBC1-dependent mechanotransduction step, as the authors seem to imply on p. 25 and elsewhere.

3) The role of co-localization seems also to be overinterpreted (p. 18, lines 377-378; p. 25, lines 518-519). By itself, co-localization of membrane proteins does not prove they physically interact, nor that physical interaction is required for downstream signaling. Many examples are known of membrane signaling proteins/channels/transporters that are closely packed, or even constitute macromolecular complexes, but appear to be relatively independent, from a functional standpoint. Once again, demonstrating physical interaction between NaBC1 and the other membrane receptors

would require considerable more work, but I do not think this hypothesis is essential for the authors' present purposes.

4) Is it possible to substitute borate with some other ion transported by NaBC1? This would provide evidence complementary to that obtained by the silencing method, and would allow to better distinguish the effect produced on the downstream signals by i) NaBC1 activation (which could directly affect other membrane proteins, as suggested by the authors), and/or ii) specific intracellular effects produced by the transported ion.

MINOR POINTS.

I noticed some confusion in the use of 'boron', 'B', 'borate' and 'borax'. For example, on p. 4 boron is defined as B, but perhaps this should be the definition of borax or borate. In fact, at the bottom of p. 5 the authors state B is an ion. On p. 27, borax is defined as boron (..hereafter..etc.), but this is not used consistently thereafter.

Throughout the paper: 'ion-channels' should read 'ion channels'.

p. 4, line 99. ..ultralow-doses.. should read ...ultralow doses...
line 101. I think '...metalloid-enhancing...' should read '...metalloid in enhancing..

p. 5, line 122. '...concentrations...are toxic..' In this case the authors are referring to solutions that are only defined later, in the Experimental Section, and boron here seems to refer to borax (see above). Is the toxic effect specifically produced by borate or could it be an effect of osmolarity at the higher concentrations used? It is not clear from the main text and the Methods section whether osmolarity correction was applied.

p. 7, line 158. '..assess..' or '..assessed..?'

p. 25, line 519. '...receptors....interacts...'

The references should be checked carefully: journal titles, capital/lower-case characters, use of abbreviations, character size, etc.

Reviewer #2 (Remarks to the Author):

Salmeron-Sánchez and colleagues perform an in-depth characterization of the role of NaBC1 in MSCs. They continue their previous work on boron to determine that boron stimulate NaBC1 and favors the osteogenic differentiation of MSCs in glass and PLLA. First, they perform an elegant experiment where they silenced NaBC1 to show that when Borax is added to the medium, NaBC1 is responsible for an increase in phosphorylated myosin expression as well as vinculin/integrins focal adhesions and cell growth. This increase in contractility and focal adhesion size correlates with an increase in SMAD and YAP expression. Interestingly, this increase in cellular contractility also correlates with an increase in osteogenic differentiation and a decrease in adipogenic differentiation. Finally, the authors show colocalization of NaBC1 with both integrins and BMPR1A. The authors propose a model where NaBC1 directly interact with BMPR1A to regulate contractile structures composed of integrins and the actin cytoskeleton. However, the authors do not show any direct interaction of NaBC1 and BMPR1A, as well as no direct interaction of integrins and the actin cytoskeleton. I suggest major revisions to address these conflicting results.

Major comments:

-Rico et al. propose a model in Figure 7 that their the data does not support. Specifically: 1) the direct interaction of NaBC1 and BMPR1A and 2) the direct link of integrins and actin. The authors should address these two assumptions or clarify them by changing the text

1. The authors perform a colocalization experiments to show colocalization of BMP1A and NaBC1 and integrins and NaBC1. To show direct interaction the authors should perform more experiments. If not, the authors should change the text. In addition, comparing the number and the size of focal adhesions at the periphery of the cell, the colocalization of NaBC1 and integrins is significantly lower and is not close to the edges of the cell.

2. Related to the direct link between integrins and actin, the authors check the role of FAK that does not seem to affect their results. With these results the authors conclude that the integrins and actin interact directly. Taking this into account the myriad of focal adhesion proteins, this seems like a huge leap. Indeed, in the same manuscript, the authors show an increase in vinculin focal adhesions with Boron. There are different vinculin-dependent mechanosensitive pathways, especially the one that involves talin too.

-The authors in a previous paper (Rico et al. 2018) showed that cellular response to NaBC1 was not dependent on integrins alpha5 beta1 and alpha5 beta3 in 3D. However, now they propose this direct link. Have the authors checked a different cellular mechanosensitive module such as the importance of membrane tension in the response that they observe? Membrane tension could lead to an increase in pMyosin, focal adhesions and YAP translocation. The authors could perform Osmotic shocks experiments or change the stiffness of the substrate. This result is important also for their model, as it could help them validate part of their model.

-The authors show that the active form of YAP does not change in In-Cell Westerns but their staining images show an overall increase in cellular YAP expression with Boron (increase in intensity). How do the authors account for this difference and for the increase in YAP expression? In fact, the images don't show a clear increase in YAP nuclear translocation as the area next to the nuclear envelope is also highly expressed. Have the authors quantified these changes? Are there differences in nuclear morphology?

Minor comments:

-The structure of the manuscript is confusing. Most of the results are related to the intercellular tension and focal adhesion growth. The differentiation results are now in the middle of the results section and I think that at the end of the manuscript would make more sense. There are also two different sections related to focal adhesions too. One potential structure could be: 1) integrins/adhesions, 2) cytoskeleton 3) YAP/SMAD, 4) differentiation.

-To observe the direct effect of Borax, a time-lapse experiment where Borax was added after some time and changes observed in cellular morphology would provide a lot of information and would add significant value to the manuscript.

Response to reviewers.

All the changes made in the manuscript have been highlighted since the main structure of the manuscript has considerably changed by suggestion of reviewer 2.

Reviewer #1 (Remarks to the Author):

The paper by Dr. Rico et al. shows that, in mesenchymal stem cells, stimulation of the NaBC1 transporter can enhance integrin-dependent cell adhesion and contractility. Moreover, stimulation promotes osteogenesis and inhibits adipogenesis. Osteogenic differentiation is found to depend on activation of the BMP pathway. The process involves co-localization of NaBC1 with integrins and BMPR1A.

Overall, the data are interesting. Elucidating the complex mechanisms linking membrane transporters to other membrane proteins in the regulation of cell adhesion, motility and differentiation, in the context of the cell's microenvironment, is an important and timely topic.

There are however several aspects of the authors' interpretation that are not fully convincing, especially concerning the first steps of the cellular mechanism (mechanosensitivity and protein-protein interaction). Demonstrating these issues would require substantial further experimentation. Alternatively, the text should be modified to remain consistent with the experimental evidence as well as clarify several unclear points.

We thank the reviewer for their supporting comments. We do agree with the observation that the demonstration of mechanosensitivity and protein-protein interaction would require further experimentation. We will modify the text to remain consistent with the experimental evidence provided.

MAJOR ISSUES.

1) The authors repeatedly refer to NaBC1 as an ion channel. However, in the presence of borate, NaBC1 actually behaves as a Na⁺-dependent borate transporter (e.g. Park et al. Mol Cell 2004). This is stated by the authors themselves (e.g. on p. 4), who also add (p. 15) that the natural ligand is borax, for which no passive diffusion is known to occur. Therefore, NaBC1 normally performs a secondary active transport, which is conceptually different from a passive flow through a conduction pore (which is typical of ion channels). Is there any evidence that these transporters have channel-like states also in the presence of borate (and not only in the absence of the latter)? If so, this should be thoroughly discussed in the paper. Otherwise, NaBC1 should not be referred to as an ion channel.

We thank the reviewer for raising this point. NaBC1 is described as a Na⁺-dependent borate co-transporter, and in absence of borate may act as an ion channel (Park et al. Mol Cell 2004).

Currently, from our knowledge, there is no evidence that these transporters have channel-like states in the presence of borate. Since our work involves the use of borax we will refer to NaBC1 as a transporter and not as an ion channel throughout the manuscript.

2) If NaBC1 is not an ion channel, it cannot be a mechanosensitive channel (Abstract and elsewhere), but could nonetheless be a mechanosensitive transporter. However, nowhere in the paper demonstration is provided that mechanical tension directly activates the transport mechanism. In the case of stretch-activated ion channels, this could be demonstrated by studying the membrane currents modulated by mechanical stimulation. Unfortunately, this procedure can be rarely applied to electrogenic transporters, whose currents have very small amplitudes. Hence, to investigate mechanotransduction in NaBC1, one should attempt to produce a suitable ad hoc experimental model (e.g. overexpressing the transporter, etc.). Another option would be to exploit the channel-like behaviour observed in the absence of borate (although this would somewhat decrease the physiological significance).

Nevertheless, I think the hypothesis that NaBC1 is a mechanosensor is not necessary for the authors' purposes. What the data show is that activation of the transporter cooperates with other membrane receptors to regulate adhesion and the contractility machinery. It is not necessary to assume an early NaBC1-dependent mechanotransduction step, as the authors seem to imply on p. 25 and elsewhere.

We thank the reviewer for these constructive comments. We do agree that demonstration of NaBC1 acts as a mechanosensitive transporter would require further experimental work. We also agree the hypothesis that NaBC1 is a mechanosensor is not essential to the manuscript. Following the reviewer's advice, we will change the text accordingly.

3) The role of co-localization seems also to be overinterpreted (p. 18, lines 377-378; p. 25, lines 518-519). By itself, co-localization of membrane proteins does not prove they physically interact, nor that physical interaction is required for downstream signaling. Many examples are known of membrane signaling proteins/channels/transporters that are closely packed, or even constitute macromolecular complexes, but appear to be relatively independent, from a functional standpoint. Once again, demonstrating physical interaction between NaBC1 and the other membrane receptors would require considerable more work, but I do not think this hypothesis is essential for the authors' present purposes.

We thank the reviewer for this subtle and insightful observation. Since the co-localization between NaBC1 and FN-binding integrins and BMPR1A does not clearly demonstrate a physical interaction occurring between these membrane receptors, and further experimental data is needed to confirm that, we will interpret our results as they are, i.e. just co-localisation of

receptors. We agree with the reviewer that this hypothesis is not essential for explaining the obtained results in the manuscript.

4) Is it possible to substitute borate with some other ion transported by NaBC1? This would provide evidence complementary to that obtained by the silencing method, and would allow to better distinguish the effect produced on the downstream signals by i) NaBC1 activation (which could directly affect other membrane proteins, as suggested by the authors), and/or ii) specific intracellular effects produced by the transported ion.

We agree with the reviewer that substituting borate with other ion would provide additional evidence and would help elucidate intracellular effects produced exclusively by the transported ion and not induced by activation of the NaBC1. However, NaBC1 transporter is quite specific for coupled Na⁺-borate transporter, thus it is not possible to substitute borate with some other similar ion. Previously, the selectivity of NaBC1 to borate was tested by measuring transport of arsenate. Boron is a metalloid, and other readily available metalloids are silicon and arsenic. Park et al; Mol Cell 2004, tested the transport of arsenate by NaBC1 showing that NaBC1 does not transport arsenate even at high concentrations (20 mM).

MINOR POINTS.

I noticed some confusion in the use of 'boron', 'B', 'borate' and 'borax'. For example, on p. 4 boron is defined as B, but perhaps this should be the definition of borax or borate. In fact, at the bottom of p. 5 the authors state B is an ion. On p. 27, borax is defined as boron (..hereafter..etc.), but this is not used consistently thereafter.

We thank the reviewer for the comment. We will use only "borax" consistently throughout the manuscript.

Throughout the paper: 'ion-channels' should read 'ion channels'.

We will change ion-channels to ion channels throughout the manuscript.

p. 4, line 99. ..ultralow-doses.. should read ...ultralow doses...

line 101. I think '...metalloid-enhancing...' should read '...metalloid in enhancing..

We will change the terms ultralow-doses to ultralow doses, and metalloid-enhancing to metalloid in enhancing as reviewer suggested.

p. 5, line 122. '...concentrations...are toxic..' In this case the authors are referring to solutions that are only defined later, in the Experimental Section, and boron here seems to refer to borax (see above). Is the toxic effect specifically produced by borate or could it be an effect of

osmolarity at the higher concentrations used? It is not clear from the main text and the Methods section whether osmolarity correction was applied.

A range of borax concentrations has been used to test cell viability as shown in Figure Supplementary S1 – this is the first figure in the manuscript. We agree with the reviewer that toxic effect due to high borax concentrations could be an effect of osmolarity. However, we did not performed additional experimental analysis to elucidate this as the higher toxic concentrations were not further used in the manuscript.

p. 7, line 158. ‘..assess..’ or ‘..assessed..’?

We will change the word assess by assessed.

p. 25, line 519. ‘...receptors....interacts...’

Corrected.

The references should be checked carefully: journal titles, capital/lower-case characters, use of abbreviations, character size, etc.

We will thoroughly check references for consistency.

Reviewer #2 (Remarks to the Author):

Salmeron-Sánchez and colleagues perform an in-depth characterization of the role of NaBC1 in MSCs. They continue their previous work on boron to determine that boron stimulate NaBC1 and favors the osteogenic differentiation of MSCs in glass and PLLA. First, they perform an elegant experiment where they silenced NaBC1 to show that when Borax is added to the medium, NaBC1 is responsible for an increase in phosphorylated myosin expression as well as vinculin/integrins focal adhesions and cell growth. This increase in contractility and focal adhesion size correlates with an increase in SMAD and YAP expression. Interestingly, this increase in cellular contractility also correlates with an increase in osteogenic differentiation and a decrease in adipogenic differentiation. Finally, the authors show colocalization of NaBC1 with both integrins and BMPR1A. The authors propose a model where NaBC1 directly interact with BMPR1A to regulate contractile structures composed of integrins and the actin cytoskeleton. However, the authors do not show any direct interaction of NaBC1 and BMPR1A, as well as no direct interaction of integrins and the actin cytoskeleton. I suggest major revisions to address these conflicting results.

Major comments:

-Rico et al. propose a model in Figure 7 that their data does not support. Specifically: 1) the direct interaction of NaBC1 and BMPR1A and 2) the direct link of integrins and actin. The authors should address these two assumptions or clarify them by changing the text.

We thank the reviewer for its helpful observation. We will clarify these assumptions changing the text along the manuscript. We will modify the proposed model without assuming physical interaction between NaBC1 and BMP1A and integrins and actin.

1. The authors perform a colocalization experiments to show colocalization of BMP1A and NaBC1 and integrins and NaBC1. To show direct interaction the authors should perform more experiments. If not, the authors should change the text. In addition, comparing the number and the size of focal adhesions at the periphery of the cell, the colocalization of NaBC1 and integrins is significantly lower and is not close to the edges of the cell.

As mentioned by both reviewer 1 and reviewer 2, the colocalization of receptors that we demonstrated does not necessarily lead to physical functional interactions between them. We will modify the text accordingly. We note that this subtle but important difference does not influence the major conclusions of the manuscript.

In relation to the position of the dots in the colocalization experiments, we did not mean that this colocalization of NaBC1 with other membrane receptors happens necessarily at focal adhesions. We will clarify this in the revised version of the manuscript.

2. Related to the direct link between integrins and actin, the authors check the role of FAK that does not seem to affect their results. With these results the authors conclude that the integrins and actin interact directly. Taking this into account the myriad of focal adhesion proteins, this seems like a huge leap. Indeed, in the same manuscript, the authors show an increase in vinculin focal adhesions with Boron. There are different vinculin-dependent mechanosensitive pathways, especially the one that involves talin too.

We agree with the reviewers that our FAK result does not mean that integrins and actin interact directly. This has been clarified in the revised version of the manuscript.

-The authors in a previous paper (Rico et al. 2018) showed that cellular response to NaBC1 was not dependent on integrins alpha5 beta1 and alpha5 beta3 in 3D. However, now they propose this direct link. Have the authors checked a different cellular mechanosensitive module such as the importance of membrane tension in the response that they observe? Membrane tension could lead to an increase in pMyosin, focal adhesions and YAP translocation. The authors could perform Osmotic shocks experiments or change the stiffness of the substrate. This result is important also for their model, as it could help them validate part of their model.

In our previous work with HUVEC cells (Rico et al; Advanced Biosystems 2018), we described that borax activation of NaBC1 in combination with VEGF enhances vascularization. We described colocalization between NaBC1 and FN-binding integrins as well as NaBC1/VEGFR. These results support the hypothesis presented in this manuscript, that NaBC1 is not a merely

boron transporter but it additionally enhances intracellular signaling. We described the same phenomenon of colocalization between NaBC1/FN-binding integrins in both works. We also demonstrated that $\alpha_5\beta_1/\alpha_v\beta_3$ integrin binding is not essential to enhance HUVEC organization in the presence of borax.

We do agree with the reviewer comment that new studies focused on measurements of membrane tension after NaBC1 activation would clarify the role of NaBC1 as a mechanosensitive transporter. Also interesting are experiments using substrates of different stiffness. These new experimental data would form the basis of new studies on the system and we will perform them in the future.

Osmotic shocks and effect of cell depolarization have been performed before by Park et al; Mol Cell 2004. Their results helped model borate transport by NaBC1 and demonstrated that NaBC1 acts as an obligated Na^+ -coupled transporter in the presence of borate and as an ion channel in the absence of borate.

-The authors show that the active form of YAP does not change in In-Cell Westerns but their staining images show an overall increase in cellular YAP expression with Boron (increase in intensity). How do the authors account for this difference and for the increase in YAP expression? In fact, the images don't show a clear increase in YAP nuclear translocation as the area next to the nuclear envelope is also highly expressed. Have the authors quantified these changes? Are there differences in nuclear morphology?

We have performed new experimental analysis to clarify the observation pointed out by reviewer concerning the inconsistencies obtained in YAP quantification.

In the revised manuscript, we have now performed image analysis to quantify nuclear YAP. The figure below shows a significant increase in nuclear YAP in the presence of borax (PLLA-B5%). We note that all our experiments quantified only active YAP. The In-Cell western data included in the original manuscript showed quantification at protein level of overall active cellular YAP, while the new figure shows quantification of just active nuclear YAP, which suggest increased translocation of active YAP to the nucleus. These new data explain discrepancies between In-Cell western and immunofluorescence quantification.

As suggested by the reviewer, we have also quantified nuclear morphology. Addition of borax reduced nuclear circularity but increased nuclear size and the nuclear aspect ratio (AR). These results suggest that activation of NaBC1 might induce intracellular tension and nuclear stress, which alters nuclear morphology. Nuclei are rounded and relaxed for cells on glass and PLLA substrates and more elongated and stressed on PLLA-B2% and PLLA-B5% substrates.

To further confirm this result, we have also quantified the amount of actin stress fibers over the nucleus. We used the nuclear outline as a mask to crop actin images and the intensity of filamentous actin in this region was quantified. The figure below shows that after NaBC1

activation, actin stress fibers over the nucleus increased. We have included representative images of nuclear morphology for cells on the different substrates, where it can be clearly seen changes in nuclear morphology as borax concentration increases.

Overall, our previous and new results suggest that NaBC1 activation increases intracellular contractility and generate intracellular tension that lead to nuclear deformation and increased presence of actin fibers in the region of the nucleus. Whether NaBC1 induces intracellular tension as a mechanosensitive protein remains to be determined, however our data suggest a novel role for NaBC1, in the activation of intracellular tension in cooperation with other membrane receptors.

Minor comments:

-The structure of the manuscript is confusing. Most of the results are related to the intercellular tension and focal adhesion growth. The differentiation results are now in the middle of the results section and I think that at the end of the manuscript would make more sense. There are

also two different sections related to focal adhesions too. One potential structure could be: 1) integrins/adhesions, 2) cytoskeleton, 3) YAP/SMAD, 4) differentiation.

We will change the structure of the manuscript to follow the reviewer suggestion.

-To observe the direct effect of Borax, a time-lapse experiment where Borax was added after some time and changes observed in cellular morphology would provide a lot of information and would add significant value to the manuscript.

[Redacted]

REVIEWERS' COMMENTS:

Reviewer #2 (Remarks to the Author):

The authors have satisfactorily addressed all my questions and suggestions as well as significantly improved the manuscript. I recommend this manuscript for publication in Communications Biology.